# Risk Factors for Intramammary Infections on Bavarian Dairy Farms—A Herd-Level Analysis

**DOI:** 10.3390/ani15172616

**Published:** 2025-09-06

**Authors:** Klara Kalverkamp, Wolfram Petzl, Ulrike S. Sorge

**Affiliations:** 1Department of Udder Health and Milk Quality, Bavarian Animal Health Services, 85586 Poing, Germany; ulrike.sorge@tgd-bayern.de; 2Clinic for Ruminants with Ambulatory and Herd Health Services, Centre for Clinical Veterinary Medicine, Ludwig Maximilians University Munich, 85764 Oberschleissheim, Germany; wpetzl@lmu.de

**Keywords:** mastitis pathogens, risk factors, dairy cattle

## Abstract

This study investigated the prevalence of mastitis pathogens and risk factors at the herd level in 305 dairy farms in Bavaria, Germany. The most common pathogen in quarter milk samples from more than 14,000 cows was identified as non-aureus staphylococci, followed by *Streptococcus uberis* and *Staphylococcus aureus*. Risk factors varied by pathogen and included factors such as herd size, type of bedding, milking system, and hygiene practices. Known preventive measures such as post-milking teat disinfection and equipment maintenance were again associated with lower infection rates. These findings highlight the importance of good management practices in reducing intramammary infections and improving udder health in dairy herds.

## 1. Introduction

Mastitis is one of the most common diseases in dairy farming globally [1] and has a profound impact on animal welfare [2]. Affected animals can show clear behavioral changes, such as shorter lying times [3,4,5], reduced rumination time and feed intake, and pinching of the tail between the hind legs, as a sign of discomfort [6].

In addition to the impact on welfare, bovine mastitis has an enormous economic effect due to milk loss [7,8,9], lower milk quality [9], treatment costs [10], and increased involuntary culling [8].

In Germany, dairy farms differ greatly between regions. In Bavaria, dairy farming has some unique characteristics that highlight the importance of regional studies. The 1,036,089 dairy cows in Bavaria alone make up 29% of all German dairy cows [11] and produce 25% of Germany’s total milk volume [12]. However, the average herd size of 44 cows is smallest in a nationwide ranking. By comparison, the herds in north-eastern Germany (Mecklenburg-Western Pomerania) are largest with an average of 244 cows per herd, and the Holstein–Friesian breed dominates the German dairy industry outside of Bavaria [13]. By contrast, the dual-purpose breed Simmental [14] is the predominant dairy breed in Bavaria, representing 77% of the Bavarian dairy cows [11]. Distinguished by their robustness, higher milk fat production, and lower culling rate, Simmental cows are growing in importance in Europe [15] and, above all, in Turkey [16]. It has also been shown that, compared to Holstein–Friesian cows, this breed has different feeding requirements in different stages of life [17]. Therefore, breed-specific factors may also impact the occurrence of IMI.

Mastitis is predominantly caused by bacterial intramammary infection (IMI), which results in local and systemic inflammation [18]. Etiologically, mastitis-causing pathogens are classified based on their main reservoir and transmission route: environmental and contagious pathogens [19]. Environmental pathogens such as *Streptococcus* (*Strep*.) *uberis* and *Escherichia* (*E*.) *coli* originate from the cows’ surroundings and may opportunistically invade the udder [20,21].

Contagious pathogens are predominantly transmitted from cow to cow and mainly include *Strep*. *agalactiae* and *Staphylococcus* (*S*.) *aureus* [22]. However, this binary classification does not fully capture the complexity of pathogen behavior; some pathogens, such as *Strep*. *dysgalactiae* [23,24,25] and the heterogeneous group of non-aureus staphylococci (NAS), can exhibit both environmental and contagious transmission characteristics [26,27].

In the mid-20th century, a five-point plan (tested milking systems, implementation of teat dipping, antibiotic treatment of clinical mastitis, antibiotic dry cow therapy, and culling of non-treatable animals) was drawn up to fight mastitis [28,29]. The main focus was on reducing IMI with contagious pathogens (*Strep. agalactiae* and *S. aureus*) and fighting infections with antibiotic dry cow therapy [30]. High somatic cell counts (SCCs) were common, and much of the emphasis was placed on reducing subclinical mastitis. The five-point plan was highly successful [30], but as a result, the prevalence of environmental pathogens has since increased, which are known to more likely cause clinical mastitis (CM) [31,32].

In this study, we focus on IMI, defined as the presence of mastitis-causing pathogens in quarter milk samples, regardless of the presence of inflammatory symptoms. There is a possibility that IMI may lead to subclinical (SCM) or CM, but it does not always result in inflammation. Therefore, although we reference studies describing risk factors for mastitis, our outcome is pathogen-based (IMI) rather than inflammation-based (SCM or CM).

The known risk factors for IMI include age and milk yield [33,34]. Cows with a higher parity are more prone to mastitis [35,36,37]. Similarly, high milk yield has been associated with an increased risk of mastitis [38,39]. Environmental causes and certain management practices have also been identified as risk factors for mastitis: Nutrition and heat stress can impair the immune competence of cows [40,41,42], while the material and hygiene of bedding materials influence cow cleanliness, which in turn affects mastitis risk [43,44,45,46]. Furthermore, poor milking hygiene and malfunctioning milking machines can increase the risk for mastitis as well. Previously described sparing factors include teat dipping [47,48], a clean calving pen [47], and a milking sequence [49]. Well-maintained milking machines [50], wearing disposable gloves, or providing feed after milking, to allow for teat closure before cows lie down, can further reduce the risk for IMI [48].

However, risk factors for IMI will vary geographically due to differences in cattle breeds, production level, and farming style, including housing, and also due to the change in prevalence since the introduction of the five-point plan.

In 2018, a Bavarian study investigated risk factors for the herd prevalence of mastitis pathogens [51], and another recent study from Bavaria analyzed the distribution of mastitis pathogens identified in the largest regional veterinary laboratory between January 2014 and December 2023 [52]. However, a re-evaluation of risk factors for IMI in this dairy region was needed, as the previous study [51] could not evaluate seasonal risk factors, and there have been legislative changes regarding antibiotic usage for livestock production in Germany and the European Union since. Specifically, in 2018, the German Veterinary Pharmacy Regulation (TÄHAV) [53] was revised, introducing an antibiogram requirement for certain cases (e.g., when changing antibiotic prescription during therapy or when using fluoroquinolones and 3rd/4th-generation cephalosporines). More recently, the National Veterinary Medicinal Products Act (TAMG) [54] was amended in 2023, extending the national antibiotic reduction program, thereby tightening monitoring and also reporting obligations for dairy cattle. Furthermore, the dairy industry has developed towards fewer but larger dairy farms and the number of farms with robotic milking is steadily increasing [55].

Therefore, the aim of this study was to describe the prevalence of the mastitis pathogens and to investigate herd-level risk factors associated with the within-herd prevalence of the common mastitis pathogens *Strep. uberis*, *Strep. dysgalactiae*, *Strep. agalactiae*, *S. aureus*, *E. coli*, and NAS in Bavarian dairy herds.

## 2. Materials and Methods

All animal-related procedures complied with institutional and national guidelines for the ethical care and use of animals. Due to the non-invasive study design, local animal ethics committee approval was not required. The participating farmers gave written informed consent for the use of their animals and voluntarily answered any questions regarding management practices.

### 2.1. Herd Selection

This cross-sectional study was conducted in Bavaria between July 2023 and July 2024. A list of all dairy Bavarian farms (*N* = 20,624) that included milk shipped daily was used to identify potential participants. Since the herd size was not available, the milk shipped was used as a surrogate for herd size. Farms with <200 kg of milk shipped daily (*n* = 2399) were excluded, as it was assumed that the herd size was fewer than 10 cows. Excluding these very small herds also increased the statistical robustness of the analysis, as estimates from very small sample sizes per herd are more prone to random variation. The remaining farms (*n* = 18,225) were divided into 4 groups based on the quartiles of daily milk haul (group 1: 201–479 kg, group 2: 480–869 kg, group 3: 870–1498 kg, group 4: ≥1499 kg). Due to logistical and financial constraints, a total of 304 herds could be examined, which were divided into groups of 76 herds per season (spring, summer, fall, and winter) for each group (1–4). The seasons were defined by month as follows: spring—March to May; summer—June to August; fall—September to November; and winter—December to February. For a good representation of the whole state of Bavaria, the herds were also divided into the six administrative districts on a percentage basis due to geographical differences in number of farms and farm sizes. Thus, 61 herds were to be studied in Franconia, 50 in Lower Bavaria, 54 in Swabia, 43 in Upper Palatinate, 52 in Upper Bavaria East, and 44 in Upper Bavaria West. The resulting farm lists were randomized (in Microsoft Excel^®^ 2019 MSO (16.101417.20007), function RAND()) and then contacted by telephone starting from the top of the list. Each farm was visited once by trained technicians from the Bavarian Animal Health Service at milking time or, in the case of robotic milking farms, at one point throughout the day.

During a farm visit, the hygiene score (1–4: 1 = clean cow, manure contamination up to dewclaws; 2 = light soiling, manure contamination up to hock joint; 3 = manure contamination above hocks and on flanks; 4 = severe manure contamination including soiled belly) [56], the hock score (1–3: 1 = no swelling or hair missing; 2 = no swelling, >1 cm diameter bald area on the hock; 3 = swelling or crust/open lesion of skin) [57], and the teat score (1–4: 1 = no hyperkeratosis; 2 = ring at teat end; 3 = moderate hyperkeratosis; 4 = extreme hyperkeratosis) [58] of lactating and dry cows were assessed on a cow level (the highest score was recorded). On farms with conventional milking systems, quarter milk samples (QMSs) were collected aseptically from all quarters of all lactating cows, prior to milking, in a sterile container with 0.5% boric acid as a preservative agent for shipment [59]. After pre-cleaning of the teats by the milker, the teat ends were rubbed with an alcohol swab and the cleanliness of the pre-cleaning was recorded (score 1–4: 1 = clean: no manure, dirt, or dip; 2 = dip present: no manure or dirt; 3 = small amount of dirt and manure present; 4 = larger amount of dirt and manure present) [56]. Finally, on conventional milking farms, the strip yield from all four quarters was manually milked in a measuring cup for 1 min (up to 10 cows per farm) immediately after cluster removal. On farms with robotic milking, animals were restrained in head locks, and aseptic QMS collection was performed independently of milking time. All samples were cooled immediately after collection and transported to the Bavarian Animal Health Service laboratory for further analysis on the same or the following day.

Based on standardized questionnaires, we recorded farm management practices, including farm type (i.e., organic or conventional), dairy and beef animal population on the farm, milking system, milking system cleaning procedures, milking processes (incl. teat dipping), milk quality metrics, dry cow management, bedding types, housing, feeding, and water supply assessments.

### 2.2. Laboratory Analysis

All QMSs were analyzed according to the guidelines of the German Veterinary Association (DVG) [60], which are based on the guidelines of the International Dairy Federation (IDF) [61] and the National Mastitis Council (NMC) [62]. An esculin agar plate with 5% sheep blood (Thermo Scientific^TM^ OXOID^TM^, Basingstoke, UK) was prepared for each cow, and an inoculum of 0.01 mL was applied to each quarter using calibrated eyelets. In the case of clinical mastitis, an additional 0.05 mL was spread on a whole agar plate. In addition to the esculin agar plate, a Sabouraud dextrose agar plate (Thermo Scientific^TM^ OXOID^TM^, Basingstoke, UK) was prepared for cows that had a history of antibiotic treatment, as yeast and fungi grow more easily on this medium. Plates were incubated at 36 ± 1 °C for 36–48 h. The analysis was performed after 36–48 h. An initial classification of the pathogens was based on colony morphology (size, color, mucus formation, and odor), hemolysis, and hemotoxin zone [60].

*S. aureus* was identified by phenotypic colony morphology and clear hemolysis with the toxin zone. If the hemolysis zone was equivocal, the clumping factor and coagulase were also tested. If negative, the colony was considered NAS. Questionable *S. aureus* colonies and all identified NASs (e.g., *S. epidermidis*, *S. chromogenes*, *S. borealis*) were examined by MALDI-TOF (MALDI Biotyper^®^ Sirius, Bruker Daltonic SPR, Hamburg, Germany).

Streptococcal differentiation was based on colony morphology, hemolysis zone, ability to cleave esculin, the CAMP test (Christie, Atkins, Munch-Petersen), and classification into Lancefield groups (B, C, G). The CAMP test with a hemolytic *S. aureus* strain was performed on all esculin-negative streptococci, but was mainly used to differentiate between *Strep. agalactiae* (CAMP test-positive) and *Strep. dysgalactiae* (CAMP test-negative) [60,62]. Hemolytic streptococci, such as *Strep. canis*, were differentiated using a commercial Lancefield group test kit (Thermo Scientific^TM^ Streptex^TM^, Nepean, ON, Canada). Esculin-positive streptococci (*Strep. uberis*, *Enterococcus* spp., *Lactococcus* spp.) were further differentiated using the selective medium kanamycin-esculin acid (KAA-Agar, produced in-house [60]) and characteristic zones of inhibition in the agar diffusion test with penicillin and rifampicin (Thermo Scientific^TM^ OXOID^TM^, Basingstoke, UK) [60]. Colonies of *Enterococcus* spp. and *Lactococcus* spp. were examined by MALDI-TOF to determine the exact species. If no species could be determined after repeated microbiological examination and MALDI-TOF, the bacteria were divided into esculin-positive and -negative streptococci according to their esculin cleavage. Due to logistical and financial constraints, not all pathogen colonies were examined using MALDI-TOF.

*Trueperella* (*T.*) *pyogenes* was also differentiated based on colony morphology and hemolysis behavior. When necessary, additional tests were performed on a Loeffler serum plate (Thermo Scientific^TM^ OXOID^TM^, Basingstoke, UK) (characteristic formation of a trench) [60] or with MALDI-TOF.

All Gram-negative bacteria (e.g., *E. coli*, *Serratia* spp., *Klebsiella* spp.) were differentiated with MALDI-TOF. Differentiation between yeasts, *Prototheca* spp., and *Norcardia* spp. was performed microscopically [60].

To classify the health status of the udder quarters, the somatic cell count (SCC) was determined for each QMS using flow cytometry (FOSSOMATIC MC, FOSS GmbH, Hamburg, Germany). The quarter was classified as healthy if the SCC was ≤100,000 cells/mL, quarters with >100,000 cells/mL were recorded as subclinical mastitis [63], and samples with visible abnormalities in the milk (e.g., flakes) were classified cow-side as clinical mastitis [19].

### 2.3. Statistical Analysis

For statistical analysis, the group classification (quartiles) was adjusted based on actually recorded cow numbers on the day of the visit rather than the initially used milk shipped daily.

SAS 9.4 software (SAS Institute Inc., Cary, NY, USA) was used for statistical analysis and alpha was set at 0.05. Descriptive statistics were calculated using PROC MEANS for continuous and PROC FREQ for categorical variables. For univariable analysis, categorical predictors were compared across continuous outcomes using Kruskal–Wallis or Mann–Whitney U-tests via PROC NPAR1WAY.

For the multivariate models, the unit of interest was the herd. Two types of outcomes were considered: For *S. aureus*, *Strep. uberis*, *Strep. dysgalactiae*, and NAS, the outcome was the within-herd prevalence. For *E. coli* and *Strep. agalactiae*, the outcome was binary (herd infected: yes/no) due to the high number of herds without the detection of these pathogens and little variation in the within-herd prevalence of positive herds.

Predictors and models were selected as follows: Initially, all potential risk factors (Table 1) were tested for an association with the prevalence of the pathogens (treated as a continuous variable) or the presence/absence of *E. coli* or *Strep. agalactiae* in a herd using appropriate non-parametric tests, such as the chi-square or Fishers exact test (PROC FREQ), Mann–Whitney U-test (PROC NPAR1WAY), or Spearman correlation (PROC CORR). If they were associated with the outcome (*p* < 0.05), they were further assessed for biological plausibility and data quality (i.e., sufficient data points per category) and collinearity. If the collinearity of potential predictors had to be assumed (e.g., r > 0.7), the more biologically plausible and data-rich variable was selected for the multivariate models. For example, the presence of *Strep. dysgalactiae* was retained instead of bulk tank somatic cell count (BTSCC), as pathogen occurrence is more likely to drive changes in SCC. Similarly, herd size was retained instead of average milk yield, as larger herds tend to have higher yields. In addition, cubicle type (deep bedded cubicles and mattress stall cubicles) was excluded when strongly correlated with the housing system, since tiestall farms by definition do not provide cubicles.

For herds infected with *E. coli* and *Strep. agalactiae*, logistic regression models were used to model the likelihood of a positive herd test with PROC GLIMMIX. For *S. aureus*, *Strep. uberis*, *Strep. dysgalactiae*, and NAS, both Poisson and negative binomial regression models were evaluated using PROC GENMOD. The natural logarithm of the herd size (i.e., number of lactating cows) was included as an offset in these models to normalize the count data (number of pathogen-positive cows per herd). In all models, the herd (either positive or negative) or number of positive cows per herd size (offset) was the unit of interest.

The predictors were selected based on aforementioned criteria, both biological and statistical, and included in the initial model. Then, backward elimination was used to achieve a parsimonious final model by stepwise removal of predictors with the highest *p*-values. Upon removal, some variables were put back into the smaller model to assess changes in the coefficients to avoid potential confounding. Model fit was assessed through residual plots and the evaluation of overdispersion. The final negative binomial model was selected over the Poisson model, because overdispersion was present and it had a lower Akaike Information Criterion (AIC).

During residual analysis of the *Strep. uberis* prevalence model, three herds were identified as strong outliers based on markedly divergent residual values that distorted the model fit. These herds were excluded from the final model to improve robustness and interpretability. As such, results for *Strep. uberis* apply to herds with similar structure and pathogen profiles. One herd with only German Yellow cattle had to be reassigned to the mixed-breed category, because it caused computational problems due to its influence on the covariate patterns in the model.

Although data were collected at the quarter and cow levels, all statistical analyses were performed at the herd level to match the study design, which focused on identifying herd-level risk factors.

## 3. Results

### 3.1. Herd Description and Farm Analysis

A total of 648 herds were contacted by telephone. The overall response rate was 47%, though the response rate increased with increasing herd size (group (G) 1 = 35%, G2 = 40%, G3 = 64%, and G4 = 75%). The most common reasons for non-participation, especially in smaller herds, were anticipated farm closure (G1 = 20%, G2 = 15%, G3 = 5%, and G4 = 4%) and “no interest in studies” (G1 = 8%, G2 = 6%, G3 = 2%, and G4 = 0%). In the end, 305 farms with a total of 14,700 lactating cows participated in this study. One additional herd was included beyond the original plan because some farmers’ responses were delayed, resulting in a total of 305 herds instead of the initially planned 304 herds. The surveys were carried out fairly evenly across herd sizes, season, and administrative district; 77 herds were surveyed in summer, 76 herds in spring, 75 herds in autumn, and 77 herds in winter. The average herd size was 48 cows (range: 9 to 250 cows). Table 2 summarizes herd characteristics and observations across the different groups. The majority of participants (83%) farmed conventionally and had Simmental cows (78% of herds). The best 25% of farms had less than 13% of cows with a hygiene score ≥ 3. Additionally, no cows on these farms had a hock score ≥ 2. However, regardless of herd size or group, on average, almost half of the cows/farm had a hygiene score ≥ 3 (39%) or had at least hairless areas on their hocks (score ≥ 2 = 44%). Yet a hock score of 3 was rarely observed (median 0%, IQR: 0–7%). At dry-off, on an average herd (median), 40% and 0% of cows per herd received antibiotic drying-off therapy (IQR: 10–100%) or internal teat sealants (IQR: 0–50%), respectively.

The BTSCC in G1 (144,000 cells/mL) was lower than that of the large herds (G3, 166,000 cells/mL, *p* = 0.04; G4: 184,000 cells/mL, *p* = 0.01; Table 2). While 95% of the G4 herds were members of the Dairy Herd Improvement Association (DHI), only 74% of small herds (G1) were. Furthermore, smaller farms used more pasture (G2: 34% vs. G4: 13%, *p* = 0.01) or tiestalls (G1: 69% vs. G4: 0%, *p* < 0.01) compared to larger herds, respectively. Additionally, the rolling herd average was significantly higher in G4 (9067 kg) herds than in smaller herds of G1 (7226 kg) through G3 (8005 kg, *p* < 0.01).

### 3.2. Prevalence of Mastitis Pathogens

A total of 58,108 QMSs were taken, of which 1.5% (*n* = 857) were contaminated (more than two pathogen types were present or were overgrown samples). Additionally, two pathogens were detected in 184 of all 57,251 QMSs. Some samples contained too little milk to assess the SCC (*n* = 1732, 3%). Of all QMSs, 88% (*n* = 50,625) tested pathogen-negative and 61% (*n* = 36,370) of samples had an SCC ≤ 100,000 cells/mL and were therefore classified as “healthy”.

The apparent prevalence is herein referred to as the prevalence. Table 3 shows the prevalence of mastitis pathogens detected in all 57,251 QMSs. While only 28% of quarters with subclinical mastitis (*n* = 18,338) were pathogen-positive, 3% of the samples classified as healthy were pathogen-positive, and 68% (*n* = 344) of the 504 samples from quarters with clinical mastitis were pathogen-positive. No clinical mastitis case could be classified as severe.

The most frequently identified group of pathogens at the quarter level were NAS, detected in 5% of all samples, followed by *Strep. uberis* (2%), *S. aureus* (2%), and *Strep. dysgalactiae* (1%). In general, the number of detections of *Strep. agalactiae* and *E. coli* was relatively low compared to that for the other pathogens. *Strep. agalactiae* and *E. coli* were detected in 0.2% (*n* = 99) and 0.1% (*n* = 69) of all QMSs, respectively.

In addition, NASs were found in 63% of all healthy pathogen-positive quarters, in 41% of all pathogen-positive QMSs from subclinical mastitis cases (*n* = 4910), and in 8% of all pathogen-positive QMSs from clinical mastitis cases. By contrast, *Strep. uberis* was the main pathogen (32%) detected in pathogen-positive QMSs with clinical mastitis (*n* = 344), followed by *S. aureus* (14%), *E. coli* (10%), and *Strep. dysgalactiae* (9%).

Table 4 shows the within-herd prevalence of selected mastitis pathogens and the number of positive herds (i.e., herds with at least one cow tested positive for the respective pathogen). Almost all herds (92%) were NAS-positive and had, on average, 13% NAS-positive cows across all groups (range: 1% to 45%).

### 3.3. Risk Factors on a Herd Level

Table 5, Table 6, Table 7, Table 8, Table 9 and Table 10 show the results of the multivariate risk factor models associated with the within-herd prevalence (*S. aureus*, *Strep. uberis*, *Strep. dysgalacatiae*, and NAS) or the odds for a herd to be positive (*E. coli* and *Strep. agalactiae*) of the various pathogens.

Herd size also influenced the within-herd prevalence of certain pathogens. While the highest within-herd prevalence of *S. aureus* was found in smaller herds (G1, *p* = 0.01), *Strep. uberis* was more frequently detected in larger herds (G4) compared to smaller herds (G1, *p* < 0.01), and *E. coli* was also more frequently detected in larger herds (G4) compared to smaller herds (G1, *p* < 0.01), based on binary infection status.

The bedding material was associated with the within-herd prevalence of *Strep. uberis*, *S. aureus*, and *Strep. dysgalactiae*. For *Strep. uberis*, bedding with straw content (lime-straw mattress, straw and lime, straw/hay) was associated with a higher within-herd prevalence of *Strep. uberis* than only mattresses with lime conditioner (*p* = 0.05), no bedding on rubber mattress stalls (*p* = 0.02), or sawdust (*p* < 0.01). By contrast, the highest within-herd prevalence of *S. aureus* or *Strep. dysgalactiae* was observed on farms with no bedding, only lime (both usually mattress stalls), or sawdust. Herds with lime-straw mattress bedding in loose bedded stalls had the lowest within-herd prevalence of those two pathogens (*p* < 0.01). In addition, organic farms had a higher within-herd prevalence of *S. aureus* (*p* < 0.01) or *Strep. uberis* (*p* < 0.01) than conventional farms.

When analyzing the milking system and milking procedure, different parameters were identified as risk factors for five of the six pathogens studied. A high level (visual assessment) of milking system hygiene was associated with a higher within-herd prevalence of *S. aureus* (*p* = 0.03) (Table 6). In addition, a maintenance contract with a milking machine company for regular maintenance of the milking equipment (5 of 11 *Strep. agalactiae*-positive herds with the contract) and also only sporadic (‘irregular’) cleaning of the water troughs (3 of 11 *Strep. agalactiae*-positive herds with irregular cleaning) were both associated with lower odds of herds being *Strep. Agalactiae*-positive (*p* = 0.04) (Table 10). Audible liner slips during milking (9 of 51 *E. coli*-positive herds) were associated with increased odds for herds being *E. coli*-positive (*p* = 0.01) (Table 9).

The within-herd prevalence of NAS (Table 8) was higher when agitated cow behavior during milking (defined as kicking or increased movement, yes/no) had been observed (*p* < 0.01). In addition, the analysis of the used post-dip components showed that chlorine dioxide-based products were associated with the lowest within-herd prevalence of NAS (*p* < 0.01). In addition, robotic milking systems had a higher within-herd prevalence of NAS compared to parlor (*p* < 0.01) or pipeline milking systems (*p* < 0.01).

The within-herd prevalence of *Strep. dysgalactiae* of herds was lower when the herd did not clean teats before milking (*p* < 0.01), when teat dips were applied post milking (*p* = 0.03), or when more cows per herd were treated with antibiotic dry-off therapy (*p* < 0.01, Table 7). In addition, the within-herd prevalence of *Strep. dysgalactiae* decreased by 1% with each 100 kg increase in rolling herd average (*p* = 0.01).

## 4. Discussion

This study examined the prevalence of mastitis pathogens and risk factors for IMI in Bavarian dairy farms at the herd level. A strength of the study was that the sample population was derived from a stratified random sampling that covered all Bavarian administrative districts, regardless of the udder health status of the herd or DHIA membership. Furthermore, an equal number of different herd sizes were visited and the study period deliberately covered all four seasons. The average herd size was 48 cows in this study, which was slightly higher than the average herd size of 44 cows in Bavaria [11]. However, this was due to the fact that farms with a daily milk yield of less than 200 kg (11.6% of all farms) were excluded from the study. The study results should therefore only be extrapolated for herds with more than 10 cows and up to 250 cows. A Bavarian study from 2018 with a similar study design had an initial list of 28,884 dairy farms in Bavaria and *n* = 24,011 farms with more than 200 kg of milk shipped daily, yet ultimately the same average herd size of 48 cows [51]. Therefore, there has been a decrease of 29% in all farms and 32% in small farms between 2017 and 2023 in Bavaria. This observation highlights the rapid changes in the Bavarian dairy industry over just a five-year span and an overproportioned loss of the smallest dairy farms of the region.

While conventional culture methods are widely used and align with current guidelines, it should be noted that molecular techniques like polymerase chain reaction (PCR) offer higher sensitivity and may improve pathogen detection, particularly for fastidious or emerging bacteria [64]. This aspect should be considered when interpreting the results.

In our study, NAS were the most frequently detected pathogens, with an apparent prevalence of 5% on a QMS level. This finding aligns with previous reports, including studies from Canada [65]. However, substantially higher NAS prevalences at the quarter level were reported in other German regions (e.g., 17% in Hesse, 9% in Brandenburg) and across Europe: Belgium—10% [66] and 33% [67]; Norway—18% [68]; Netherlands—11% [69]; or Finland—11% [70] and 43% [71]. This can largely be attributed to differences in study design, sampling criteria, or diagnostic methods. For instance, the highest values came from studies that used PCR or repeatedly sampled healthy herds [67,71].

An evaluation of laboratory submissions from Bavaria, conducted by Bechtold et al. (2024), showed an increase in NAS detection from 25% (2014) to 35% (2023) within pathogen-positive QMSs [52]. That sample included submissions from individual mastitic cows and we found NAS most likely in subclinical cases. However, a previous Bavarian study with a similar unbiased study design reported 4% NAS at the QMS level [51]. Since that value is numerically lower than our results, this may indicate a slight upward trend in prevalence. The widespread and increasing detection of NAS across studies may point to their growing relevance in dairy production. In our study, automatic milking systems were associated with a higher prevalence of NAS on the herd level. Automatic milking systems have become increasingly prevalent. NAS occur naturally on the cow’s skin and teat canal and may infect the udder, if the milking hygiene [26,72] or the milking process is suboptimal, which puts strain on the teat canal. For instance, agitated cows during milking were associated with a higher within-herd prevalence in this study. The higher milking frequency of automatic milking systems compared to the traditional 2x/d milking [73] might also affect their presence in the teat canal [74]. While beyond the scope of this cross-sectional study, it is noteworthy that, compared to Groh et al., 2023 [51], a higher proportion of farms in our dataset already used automatic milking systems (9% vs. 21%), reflecting the general trend towards increasing technology use in dairy farming. In this context, future research may explore how precision livestock farming tools could contribute to earlier detection and improved prevention of mastitis. This aligns with recent findings by Lavrijsen-Kromwijk et al. (2024) [55], who reported that German dairy farms implementing more technology benefited from reduced contamination and lower lameness prevalence.

The choice of teat dip was also important for the within-herd prevalence of NAS. Similar to the reports by Oliver et al. (1989) [75], we demonstrated that chlorine dioxide was associated with lower NAS prevalence on a herd level. This was not seen with the other pathogens evaluated under this study. Given the heterogeneous species composition of “NAS” and variable pathogenic potential [26], it should be noted that NAS are discussed as a group in this study, and not as individual species, despite the known differences in pathogenicity between species. Further research is warranted to better understand their epidemiology and develop targeted control strategies, if NASs become problematic for the udder health of a herd.

In this study, the most frequently isolated pathogens (after NAS) were *Strep. uberis* and *S. aureus*, with QMS prevalences of 1.9% and 1.8%, respectively. *Strep. dysgalactiae*, ranked fourth with a QMS prevalence of 1%, which was comparable to other studies from Germany [76]. Among these, *Strep*. uberis showed the strongest association with clinical mastitis in this study, a finding supported by previous research [46,77]. A very small proportion of isolates (< 0.1% of all QMSs) could not be definitively identified beyond esculin-positive or -negative streptococci despite repeated microbiological testing, and are therefore unlikely to substantially affect the reported prevalence of specific pathogens.

Compared to a previous Bavarian study [51], the prevalence of *S. aureus* decreased (from 3% to 2%), while that of *Strep. uberis* increased (from 1% to 2%) and that of *Strep. dysgalactiae* remained the same (1% to 1%) at the QMS level. This shift was also reflected in a 10-year retrospective analysis of positive QMSs, where *S. aureus* decreased from 25% (2014) to 16% (2023) and *Strep. uberis* increased from 17% to 22% [52]. Together, these findings indicate a gradual transition from contagious to environmental pathogen dominance. Studies from other regions showed wider variations. In Hesse, *Strep. uberis* was more prevalent (9%) than *S. aureus* (5%) at the QMS level [78], probably due to the inclusion of herds with known udder health problems. By contrast, the Brandenburg study reported a higher *S. aureus* prevalence (6%) and lower *Strep. uberis* prevalence (1%) at the QMS level [76], possibly due to the focus on clinically healthy animals. Similar to our study, a Dutch study showed a decrease in *S. aureus* from 6% (1973) to 2% (2003), while *Strep. uberis* remained fairly stable (1–2%) [69]. A higher *S. aureus* quarter-level prevalence was observed in Norway (21%) and Finland (up to 21%) compared to *Strep. uberis* (7–9%) [68,70,71]. Methodological factors, regional antimicrobial policies, breed, nutrition, or climate differences likely contribute to these differences.

At the herd level, the comparatively high within-herd prevalence of *Strep. uberis* and *S. aureus* on organic compared to conventional farms was noteworthy (*p* < 0.01). Groh et al. (2023) [51] was previously unable to find an association between organic farming and an increased prevalence of either *Strep. uberis* or *S. aureus* in the same region. However, *S. aureus* had been previously reported as more prevalent in organic farms than in conventional farms [79] in other parts of the world. This might be because organic farmers refrain from treating cows with subclinical mastitis with antibiotics in order to avoid economic strain due to the long withdrawal times or even lose the animal to organic production in accordance with the limits set by their respective organic standards (maximum number of antibiotic treatments per dairy cow per year to obtain organic certification: three times in the European Union and none in the United States) [80,81]. Therefore, by preventing cures, they are increasing the prevalence within their herds. A higher level of dry cow therapy (the more cows per herd, %) was associated with a lower within-herd prevalence of *Strep. dysgalactiae* (*p* < 0.01), which was also previously reported by Groh et al. (2023) [51]. However, because the analysis was conducted at the herd level, no conclusions can be drawn about the treatment status of individual infected cows and, therefore, the potential for an ecological fallacy has to be acknowledged. Nevertheless, the observed association underlined the importance of effective dry cow protocols.

Housing, especially bedding, was another important factor for these three pathogens. Several studies have described straw as a reservoir for *Strep. uberis* [43,44,82,83]. According to European regulation 2018/848 for organic farming, bedding is mandatory on organic farms [80]. While Barth et al. (2010) reported that straw was always a part of the bedding on German organic farms [84], we did not see this. In our study, straw was a bedding component on 81% of organic farms (conventional farms: 60%; *p* < 0.01). However, this might still drive our observation, where *Strep. uberis* was more prevalent in organic compared to conventional farms. Furthermore, we observed that the absence of bedding (e.g., lime use only) or the use of rubber mats as lying surfaces increased the within-herd prevalence of *S. aureus* and *Strep. dysgalactiae* (*p* < 0.01). A study from Southern Ethiopia also found that no litter was associated with an increased detection rate of *S. aureus* [49]. It is known that *S. aureus* is not only detectable within the mammary gland, but also on the skin or in the environment [85,86]. Leaked milk from infected quarters might contain enough viable *S. aureus* for some time to infect the next cow that uses that cubicle, as *S. aureus* easily survives on these surfaces [87]. Similarly, *Strep. dysgalactiae* has to be considered environmental as well as contagious [23]. Therefore, regular changes or adding of fresh bedding will likely decrease the bacterial load on the lying surface and therefore reduce the risk of transmission between cows.

Furthermore, in this study, the absence of teat cleaning was associated with a lower *Strep. dysgalactiae* within-herd prevalence. One might argue that well-maintained cubicles resulted in better cow and udder hygiene where farmers may not see the need to clean teats prior to milking. This assumption is supported by previous findings where the hygiene of cows and their udders is greatly influenced by the cleanliness of their lying areas [88]. Comparable to our study, the use of a post-dip has been previously identified as a protective factor towards *Strep. dysgalactiae* IMI [33,42,51]. Interestingly, the *Strep. dysgalactiae* within-herd prevalence decreased with increasing herd performance in this study. Again, one could argue that higher milk yields may be accomplished because of better herd management practices [89].

In addition, a clean milking system was associated with a higher *S. aureus* within-herd prevalence (*p* = 0.03). One might speculate that *S. aureus* infections were first observed on the farm and then more emphasis was placed on a clean milking system. This interpretation is consistent with findings from sanitation programs for *S. aureus*-positive herds, where enhanced milking hygiene and strict milking routines are key components of control strategies [90]. While this study identifies associations between various factors and pathogen prevalence, it is important to note that the cross-sectional design does not allow for causal interferences. Longitudinal studies are needed to explore causal relationships more thoroughly.

Furthermore, it is noteworthy that *Strep. uberis* was detected more frequently in herds with Brown Swiss cattle. Breed has already been described as a risk factor [91,92]. Brown Swiss cattle have a fundamentally different immune response than Holstein Friesian animals and should therefore be more resistant to IMI than other breeds [93]. However, we found the opposite at the herd level and future studies need to further evaluate this. One possible explanation could be seasonal influences: Brown Swiss cows have been reported to show autumn low milk yield syndrome, potentially related to summer heat stress, which was also associated with increased mastitis occurrence [94]. Another aspect might be breed-specific udder and teat morphology. Brown Swiss cows are characterized by longer and thicker teats [95], which could potentially influence milking characteristics and pathogen transmission dynamics. While our cross-sectional design does not allow conclusions to be made on causality, such breed-related physiological and morphological traits may partly contribute to the observed associations and should be investigated in more detail in future studies.

The quarter-level prevalence of *E. coli* and *Strep. agalactiae* remained very low (0.1% and 0.2%, respectively), which is consistent with previous findings in the region [51]. Long term data showed a downward trend of *Strep. agalactiae* prevalence in Bavaria (5% to 3%) [52], Finland (0.12% to 0.02%) [70], or the Netherlands (7% to 0%) at the QMS level [69]. Although both pathogens were so rarely found, and we had to use a different statistical approach, we were still able to find statistical differences despite a potentially low power.

For instance, *Strep. agalactiae* was detected in only 11 out of 305 herds, and only 3 of these herds reported irregular cleaning of water troughs. Despite the low number of *Strep. agalactiae*-positive herds, we could show that, contrary to the assumption that *Strep. agalactiae* is a classical contagious pathogen, the cleaning of water troughs was associated with a reduced likelihood for a herd to be *Strep. agalactiae*-positive. Since a recent Norwegian study found *Strep. agalactiae* in the environment and especially in water troughs [96], the environmental component of *Strep. agalactiae* infections needs to be acknowledged. At the same time, the contagious nature of this pathogen was confirmed, as the regular maintenance of milking equipment was a protective factor in this study. The original five-point plan to combat contagious mastitis pathogens emphasized the importance of well-set and well-maintained milking equipment. This shows that these findings are still relevant and that careful maintenance of milking equipment remains essential to good udder health.

Similarly to *Strep. agalactiae*, the median within-herd prevalence of *E. coli* was very low (0%) and the overall herd-level prevalence of *E. coli* was also low (only 51 positive herds). Due to the short residence time of *E. coli* in the udder, the likelihood of a cow in a large herd testing positive for *E. coli* is higher compared to a cow in a smaller herd at any given day, as the total number of animals tested is higher. Therefore, larger herds are more prone to test positive for *E. coli*, and the lower likelihood of smaller herds to be *E. coli*-positive should therefore not be overinterpreted. This likely resulted in the higher detection risk of *E. coli* in our study compared to Groh et al. (2023) [51], because this study included more large herds (>100 cows: 22 vs. 2) than the previous one.

The National Mastitis Council previously reported a link between liner slips and environmental mastitis, but without naming specific pathogens [97]. In this study, we could show an association between the presence of audible liner slips and the *E. coli* status of herds. As *E. coli* can occur as an environmental pathogen both on liners and on the teat skin, the sudden influx of air during a liner slip could cause milk particles and pathogens to return into the udder. The study was therefore able to confirm previously named risk factors but also identified new risk factors, such as the lack of bedding for *S. aureus* and *Strep. dysgalactiae.* It is worth noting that none of the within-herd prevalences or odds for a positive herd were associated with season in this study. This was surprising as Bechtold et al. (2024) [52] found more environmental pathogens in quarter-milk samples during the summer months in the same region, and cows shed *Strep. uberis* more during the hot season [40]. However, while the bulk tank SCC also rises during the summer in Bavaria [98], we were unable to find a seasonal effect on the within-herd prevalence or odds for positive herds of environmental or other pathogens. Seasonal influences, such as heat or humidity, often affect individual cows, but they may not result in a measurable shift in overall within-herd prevalence. Potentially, this is an individual cow problem as the samples of Bechtold et al. (2024) included approximately 20–30% of individual cow submissions by farmers or veterinarians. By contrast, this study sampled the entire milking herd regardless of disease status.

Despite its broad scope, this study had several limitations. The cross-sectional design did not allow temporal effects to be measured, such as short-term fluctuations in bacterial shedding or changes over time. In addition, the number of positive herds was relatively small for some pathogens, such as *E. coli* and *Strep. agalactiae*, which limited the statistical power of their risk factor analysis.

## 5. Conclusions

This study provides a comprehensive overview of IMI in Bavarian dairy herds and identifies key management factors at the herd level associated with their occurrence. By systematically sampling all lactating cows from randomly selected herds, the study provides an unbiased view of the presence of relevant mastitis pathogens on small- to mid-sized farms. Frequent pathogens, such as *S. aureus*, *Strep. uberis*, and *Strep. dysgalactiae*, were analyzed based on their within-herd prevalence. Less frequent pathogens, such as *E. coli* and *Strep. agalactiae*, were evaluated based on their herd-level presence. The results underscore the importance of adequate bedding, dry-off treatment, and milking hygiene in minimizing pathogen presence. These findings provide a basis for developing pathogen-specific prevention strategies that are feasible and relevant for small- and medium-sized dairy herds.

## Figures and Tables

**Table 1 animals-15-02616-t001:** Evaluated risk factors for the within-herd prevalence of *Strep. uberis*, *S. aureus*, *Strep. dysgalactiae*, *Strep. agalactiae*, *E. coli*, and NAS.

Parameter	Variable
Herd Size	Group 1 = 9–29 cows, Group 2 = 30–47 cows, Group 3 = 48–69 cows, Group 4 = ≥70 cows
Season	Spring, Summer, Fall, Winter
Farm- and herd structure	Farm organization (*organic, conventional*), breed (*Simmental, Brown Swiss, Holstein, German Yellow cattle, mixed*), DHI ^1^, farm type (*dairy farming, young stock raising, bull fattening, field crop, biodigester, other*), open farm ^1^ (*biocontrol of purchased animals* ^1^)
Udder health	Rolling herd average (kg), bulk tank somatic cell count, bulk tank bacteria count, flaming of the udder hair, trimming of tail tassel, frequency of hoof care per year, hygiene score ^2^, teat end score ^2^, and hock score ^3^ of adult cows
Milking and milking system	Milking (*robotic milking, milking parlor, pipe milking system, other* ^5^), liner (*rubber, silicone*), maintenance agreement ^1^, system service (*regular, only when required*), number of milking units Hygiene of milking system, hygiene milking clusters, hygiene milk filter, frequency of milk filter change Automatic cluster removal ^1^, machine stripping (*never, automatic, manual by pushing the claw down*), cluster position ^1^, audible liner slips ^1^, disposable gloves ^1^, overmilking (*never, at start or end of milking, both*), milking sequence ^1^, pre-stripping (*never, with pre-milk cup, without pre-milk cup*), restless cows during milking ^1^, teat cleaning (*not done, dry, moist, with pre-dip*), post-dip ^1^, dip coverage (*<50% of teat, ≥50% of teat*), dip agent category (*iodine, lactic acid, chlorhexidine, chlorine dioxide, other* ^6^), intermediate cluster disinfection (ICD) (*none, automatic, manual*), ICD-type (*peracetic acid, steam, other* ^7^) Milking robot: attachment works without problems ^1^, selection pen ^1^, exit protected ^1^, fetched cows/day (%), manual cleaning of exterior of the robot (times/day), interior rinse (minutes after last cow), main cleaning cycles (times/day)
Feeding	Head locks ^1^, clean feeding table ^1^, rough feeding table surface ^1^, fresh food (n/day), push up of feed (n/day), cattle sort feed ^1^, assessment of feed (*no feed on feed bunk at visit, insufficient, good*) Total mixed ration (TMR) ^1^, partial TMR ^1^, hay ^1^, fresh cut greens ^1^, minerals ^1^, wet feedstuffs (*none, spent grains, wet pulp, mix*), feed stuff analysis (frequency/year), transit feed ration (*yes, no, anionic salts*)
Water	Water source (*municipal, tested or untested well*), regular cleaning of water troughs ^1^, hygiene of water trough (*clean, slightly soiled, severely soiled*), sufficient number of water throughs (>9 cm/cow) ^1^, adequate water flow (>15 L/min) ^1^
Housing	Housing (*freestall, tiestall*), mattress or deep bedded cubicles, calving pen (*yes, no, also used as sick pen*), pasture-access ^1^, outdoor pen ^1^, ventilators ^1^, correctly lying ^1^, unobstructed standing up and lying down ^1^, robotic manure scraper ^1^, frequency/day of manure scraper, overcrowding ^1^, bedding (*none, lime, straw with lime, lime-straw mattress, straw or hay, sawdust, recycled manure solids (incl. biodigester substrate, other* ^8^))
Dry cow management	Type of dry-off (*abrupt, intermittent, other* ^9^) at dry-off ^4^: CMT, milk samples for bacteriological determination, antimicrobial treatment during lactation at the end of lactation, antibiotic dry-off therapy, internal teat sealant, bolus, homeopathy

^1^ Yes or no; ^2^ % of score ≥3; ^3^ % of score ≥2; ^4^ Cow/herd; ^5^ Rotary milking system, bucket milking machine; ^6^ Effective microorganisms, tea tree oil, hydrogen peroxide; ^7^ Hydrogen peroxide, water; ^8^ Fermentation substrates of plant and/or animal origin and manure from other animals; ^9^ Farm-specific dry-off.

**Table 2 animals-15-02616-t002:** Description of the herds and farms for 305 Bavarian dairy farms. All scores and husbandry refer to lactating cows.

Parameter	Group 1	Group 2	Group 3	Group 4	Overall
*N*	77	75	76	77	305
Herd size ^1^, *n*					
Total dairy cows	22 (19–25)	35 (32–40)	56 (51–63)	87 (75–125)	48 (29–70)
Dry cows	2 (1–3)	4 (2–5)	6 (5–9)	10 (7–15)	5 (3–8)
Rolling herd average milk, kg	7226 (5854–8198)	7728 (6552–8793)	8005 (7085–8955)	9067 (7900–9897)	7974 (6991–9076)
Bulk tank bacteria count (10^3^/mL)	13 (9–20)	14 (8–25)	11 (8–20)	14 (9–20)	13 (9–20)
Bulk tank somatic cell count (10^3^/mL)	144 (100–207)	168 (111–235)	166 (123–217)	184 (130–220)	168 (117–220)
Organic production, %	13	27	17	12	17
Member dairy herd improvement association, %	74	81	91	95	85
Breed, %					
Simmental	86	74	75	78	78
Brown Swiss	7	7	11	5	7
Holstein	0	0	3	4	2
other ^2^	8	19	12	13	13
Hygiene score ≥ 3 ^3^, %	38 (21–75)	44 (14–77)	40 (14–64)	31 (10–60)	39 (13–70)
Hock score ≥ 2 ^4^, %	46 (17–95)	35 (10–91)	49 (19–97)	36 (17–96)	44 (15–95)
Teat cleanliness score ≥ 3 ^5^, %	23 (8–50)	25 (10–66)	36 (10–63)	40 (18–68)	30 (10–60)
Hyperkeratosis score ≥ 3 ^6^, % cows/herd	0 (0–4)	0 (0–5)	0 (0–6)	0 (0–2)	0 (0–4)
Milking system, %					
Milking parlor	30	51	63	51	49
Pipe milking system	68	36	9	0	28
Robotic milking system	1	12	26	46	21
Other ^7^	1	1	1	4	2
Housing, %					
Freestall	35	67	93	100	74
Tiestall	65	33	7	0	26
Mattress stalls cubicles	18	28	49	40	34
Deep bedded cubicles	13	32	49	61	39
Pasture	26	34	22	13	23
Dry cow management, % cows/herd					
Internal teat sealant	0 (0–0)	0 (0–30)	0 (0–90)	0 (0–100)	0 (0–50)
Antibiotic dry-off	50 (8–100)	30 (0–90)	37 (13–90)	50 (20–100)	40 (10–100)

^1^ Median (25th–75th percentile); ^2^ mixed herds with different breeds (G1: *n* = 6, G2: *n* = 15, G3: *n* = 8, G4: *n* = 10), German Yellow cattle (G3: *n* = 1) (overall: *n* = 40); ^3^ Score according to Cook and Reinemann; ^4^ score according to University Cornell; ^5^ score according to Cook and Reinemann; ^6^ Score according to NMC; ^7^ rotary milking system (G4: *n* = 2) or bucket milking machine (G1: *n* = 1, G2: *n* = 1, G3: *n* = 1, G4: *n* = 1).

**Table 3 animals-15-02616-t003:** Apparent prevalences of mastitis pathogens from all quarter milk samples without contamination (*n* = 57,251), including no-growth samples (*n* = 50,625) and divided, based on SCC on quarter level, into healthy (≤100,000 cells/mL, *n* = 36,370), subclinical mastitis (>100,000 cells/mL, *n* = 18,338), and clinical mastitis (*n* = 504) samples. SCC could not be assessed in 1732 samples (3%), due to insufficient milk volume.

	All Quarter Milk Samples (*n* = 57,251)		Pathogen-Positive			
			Overall (*n* = 6625)	Healthy (*n* = 1269)	Subclinical (*n* = 4910)	Clinical (*n* = 344)
Pathogen	*n*	%	%	%	%	%
Non-aureus Staphylococci (NAS)	2847	5.0	42.9	62.9	40.6	8.4
*Streptococcus uberis*	1062	1.9	16.0	5.3	19.4	32.3
*Staphylococcus aureus*	1020	1.8	15.4	20.4	13.6	14.0
*Streptococcus dysgalactiae*	521	0.9	7.9	2.7	8.9	8.7
*Lactococcus garviae*	265	0.5	4.0	2.1	4.4	1.7
*Enterococcus faecalis*	235	0.4	3.5	2.0	4.0	1.2
*Streptococcus agalactiae*	99	0.2	0.7	0.5	1.8	0.3
*Serratia* spp.	92	0.2	1.5	0.1	1.2	6.4
*Lactococcus lactis*	80	0.2	1.4	0.8	1.3	0.9
*Trueperella pyogenes*	78	0.1	1.2	1.7	0.8	5.2
*Escherichia coli*	69	0.1	1.2	0.1	0.8	10.2
Other ^1^	44	0.1	1.0	0.2	0.2	2.3
*Citrobacter* spp.	42	0.1	0.6	0.3	0.7	0.6
Other esculin-pos. Streptococci	33	0.1	0.5	0.3	0.3	0.9
*Enterococcus* spp.	31	0.1	0.5	0.2	0.5	0.3
*Klebsiella* spp.	26	<0.1	0.4	0.1	0.4	2.0
Yeast	26	<0.1	0.4	0	0.4	2.0
*Streptococcus gallolyticus*	24	<0.1	0.4	0.3	0.4	0.3
Other esculin-neg. Streptococci	14	<0.1	0.2	0.2	0.1	1.7
*Prototheca* spp.	8	<0.1	0.1	0.1	0.1	0.6
*Streptococcus canis*	5	<0.1	0.1	<0.1	<0.1	<0.1

^1^*Streptococcus pluranimalium*, *Weissella cibaria*, coryneform bacteria, *Norcadia* spp., *Streptococcus hyovaginalis*, *Enterobacteriaceae*, *Helococcus ovis*, *Streptococcus lutetiensis*, *Proteus* spp. *Pasteurella multocida*, *Pseudomonas aeruginosa*, *Enterobacter* spp., *Aerococcus* spp.

**Table 4 animals-15-02616-t004:** Within-herd prevalence of the mastitis pathogens by group (median, 25th and 75th percentiles) as well as percent-positive herds, defined as herds with at least one cow with the respective pathogen.

Pathogen	Within-Herd Prevalence (%)					Herds Positive *n* (%)
	Group 1	Group 2	Group 3	Group 4	All	All
Herd size, range of cows	9–29	30–47	48–69	70–250	9–250	
Non-aureus Staphylococci (NAS)	11 (0–19)	13 (7–19)	13 (8–20)	14 (9–24)	13 (7–20)	279 (92)
*Staphylococcus aureus*	4 (0–11)	4 (0–10)	2 (0–6)	1 (1–3)	3 (0–8)	211 (69)
*Streptococcus uberis*	0 (0–6)	3 (0–7)	3 (0–7)	4 (2–8)	3 (0–7)	205 (67)
*Streptococcus dysgalactiae*	0 (0–5)	2 (0–5)	2 (0–4)	1 (0–3)	2 (0–4)	173 (57)
*Streptococcus agalactiae*	0 (0–0)	0 (0–0)	0 (0–0)	0 (0–0)	0 (0–0)	11 (4)
*Escherichia coli*	0 (0–0)	0 (0–0)	0 (0–0)	0 (0–1)	0 (0–0)	51 (17)

**Table 5 animals-15-02616-t005:** Negative binomial model to assess risk factors associated with *Strep. uberis* within-herd prevalence. Model excludes 3 herds that were severe outliers.

Parameter		Prevalence	95% CL		*p*-Value
	*n*	Ratio	Lower	Upper	
Intercept		−4.09	−5.04	−3.12	<0.01
Breed					
Brown Swiss	22	3.01	1.12	7.85	0.02
Simmental	238	1.86	0.73	4.52	0.17
Mix	38	2.05	0.79	5.14	0.13
Holstein	5	Reference			
Production system					
Organic	52	1.58	1.17	2.15	<0.01
Conventional	253	Reference			
Bedding ^1^					
Recycled manure solids	9	0.87	0.41	1.86	0.72
Lime	24	1.04	0.60	1.81	0.88
Lime-straw mattress	44	1.24	0.52	2.00	0.45
Straw with lime	57	1.70	1.06	2.60	0.02
Straw	92	1.41	0.92	2.12	0.12
Sawdust	38	0.69	0.39	1.19	0.17
Mattress cubicle, none	41	Reference			
Dispersion		0.52	0.36	0.73	

^1^ Recycled manure solids included fermentation substrates (biodigester) of plant and/or animal origin and manure from other animals. A ‘lime-straw mattress’ refers to a premixed mattress of lime and chopped straw for cubicles. The term ‘straw and lime’ refers to separately added straw and lime, usually by hand, to the cubicle. The term ‘straw/hay bedding’ refers to either pure straw, pure hay, or a mixture of both.

**Table 6 animals-15-02616-t006:** Results of the negative binomial model to identify risk factors associated with the within-herd prevalence of *S. aureus*.

Parameter		Prevalence	95% CL		*p*-Value
	*n*	Ratio	Lower	Upper	
Intercept		−3.59	−4.19	−2.98	<0.01
Group					
1	77	2.93	1.77	4.82	<0.01
2	75	2.09	1.26	3.45	<0.01
3	76	2.01	1.21	3.30	<0.01
4	77	Reference			
Production system					
Organic	52	2.05	1.37	3.12	<0.01
Conventional	253	Reference			
Hygiene milking system					
Visibly clean	199	1.55	1.08	2.22	0.02
Visibly soiled	105	Reference			
Bedding					
Recycled manure solids	9	0.62	0.23	1.83	0.36
Lime-only	24	0.64	0.31	1.36	0.23
Lime-straw mattress	44	0.26	0.15	0.54	<0.01
Straw with lime	57	0.48	0.26	0.85	0.01
Straw	92	0.73	0.43	1.21	0.23
Sawdust	38	0.85	0.47	1.55	0.60
Mattress cubicle, none	41	Reference			
Dispersion		1.01	0.76	1.34	

**Table 7 animals-15-02616-t007:** Results of the negative binomial model to identify risk factors associated with the within-herd prevalence of *Strep. dysgalactiae*.

Parameter		Prevalence	95% CL		*p*-Value
	*n*	Ratio	Lower	Upper	
Intercept		−2.57	−3.12	−2.03	<0.01
Teat cleaning					
Moist	136	1.05	0.79	1.41	0.73
None	19	0.43	0.22	0.86	0.02
Predip ^1^	28	1.33	0.82	2.16	0.25
Dry	122	Reference			
Application of post-dip					
No	129	1.36	1.03	1.79	0.03
Yes	176	Reference			
Bedding					
Recycled manure solids	9	0.84	0.39	1.80	0.65
Lime	24	0.96	0.57	1.62	0.87
Lime-straw mattress	44	0.21	0.11	0.39	<0.01
Straw with lime	57	0.70	0.45	1.07	0.05
Straw	92	0.57	0.38	0.87	0.01
Sawdust	38	0.77	0.46	1.28	0.31
Mattress cubicle, none	41	Reference			
Antibiotic dry-off ^2^		0.99	0.98	0.99	<0.01
Rolling herd average, kg		0.99	0.98	0.99	0.01
Dispersion		0.45	0.27	0.75	

^1^ Disinfected wipes; ^2^ cows/herd.

**Table 8 animals-15-02616-t008:** Results of the negative binomial model to identify risk factors associated with the within-herd prevalence of NAS.

Parameter		Prevalence	95% CL		*p*-Value
	*n*	Ratio	Lower	Upper	
Intercept		−2.36	−2.82	−1.90	<0.01
Milking system					
Robotic milking system	65	1.74	1.36	2.14	<0.01
Milking parlor	153	1.12	0.91	1.37	0.28
Pipe milking system	87	Reference			
Dip agent base					
Chlorine dioxide	3	0.24	0.10	0.60	<0.01
Chlorhexidine	20	0.98	0.65	1.48	0.94
Iodine	87	0.81	0.58	1.12	0.21
Other	15	0.85	0.61	1.18	0.33
Lactic acid	51	0.96	0.67	1.36	0.80
No post-dip used	129	Reference			
Agitated cows during milking					
Yes	42	1.34	1.09	1.66	<0.01
No	257	Reference			
Dispersion		0.23	0.17	0.31	

**Table 9 animals-15-02616-t009:** Results of the logistic regression model to identify risk factors associated with *E. coli*-positive herds.

Parameter		Odds	95% CL		*p*-Value
	*n*	Ratio	Lower	Upper	
Intercept (estimate)		0.59			
Group					
1	77	0.03	0.003	0.26	<0.01
2	75	0.21	0.06	0.69	0.01
3	76	0.51	0.17	1.45	0.20
4	77	Reference			
Audible liner slips					
No	202	0.16	0.05	0.45	<0.01
Yes	32	Reference			

**Table 10 animals-15-02616-t010:** Results of the logistic regression model to identify risk factors associated with *Strep. agalactiae*-positive herds.

Parameter		Odds	95% CL		*p*-Value
	*n*	Ratio	Lower	Upper	
Intercept (estimate)		−2.67			<0.01
Maintenance contract, milking system					
No	237	0.28	0.08	0.97	0.04
Yes	67	Reference			
Water trough cleaning					
Irregularly	31	4.36	1.04	18.34	0.04
Regularly	274	Reference			

## Data Availability

Data are contained within the article.

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
