# Peer review of "Risk Factors for Intramammary Infections on Bavarian Dairy Farms—A Herd-Level Analysis"

_animals, 2025, doi:10.3390/ani15172616_

Round 1
Reviewer 1 Report
Comments and Suggestions for Authors
This manuscript investigates farm-level factors associated with the presence of key mastitis-causing bacteria on German dairy farms. It is an analytical observational study in which the authors collected data from just over 300 herds through questionnaires, on-farm observations, and milk sample testing. Statistical models were then used to examine how management, housing, and milking practices relate to the occurrence of each pathogen. The strength of this work lies in its practical relevance. Mastitis remains one of the most costly and challenging health issues in dairy production, and identifying pathogen-specific risk profiles can help farmers and veterinarians develop targeted prevention strategies.
Comments for the Authors:
1- The Simple Summary and Abstract state 305 herds; the Results also mention 305 enrolled; however, the Herd Selection section in Methods indicates that 304 herds “could be examined.” Please clarify whether 304 was the intended sample or the number actually examined, and make this consistent throughout the manuscript.
2- The “Laboratory analysis” section describes several criteria and that should be supported with references to standard protocols or relevant previous studies.
3- Please define AMS in the Table 2 in a footnote.
Comments on the Quality of English Language
The manuscript would benefit from careful proofreading. For example:
Line 430: “Since that is numerically lower than our results and it also may indicate a slight upward trend” is incomplete, as “Since” introduces a dependent clause without a corresponding main clause. Consider rephrasing for clarity.
Table 2: Correct “Teat clealiness Score” to “Teat cleanliness Score” and “Deep beeded Cubicles” to “Deep bedded cubicles.”
The sentence across lines 447–449 contains a long sequence of pathogen names, prevalence values, and rankings, which makes it heavy to read. Consider rephrasing to improve flow and readability.
Author Response
Thank you for carefully reviewing our manuscript and providing constructive and insightful feedback. We greatly appreciate the time and effort you dedicated to evaluating our work. To enhance clarity and readability, MDPI Author Services professionally proofread the manuscript, and we reformatted all tables for consistency and visual clarity. All changes made in response to your comments are highlighted in yellow in the revised manuscript. Our point-by-point responses to your comments are provided below.
This manuscript investigates farm-level factors associated with the presence of key mastitis-causing bacteria on German dairy farms. It is an analytical observational study in which the authors collected data from just over 300 herds through questionnaires, on-farm observations, and milk sample testing. Statistical models were then used to examine how management, housing, and milking practices relate to the occurrence of each pathogen. The strength of this work lies in its practical relevance. Mastitis remains one of the most costly and challenging health issues in dairy production, and identifying pathogen-specific risk profiles can help farmers and veterinarians develop targeted prevention strategies.
- AU: We sincerely thank you for carefully reading our manuscript and for providing such a clear and thoughtful summary of our study. We truly appreciate that you recognized the practical relevance of our work and emphasized the importance of identifying pathogen-specific risk profiles for mastitis prevention. Your acknowledgement of the study’s approach and potential contribution to farmers and veterinarians is very encouraging for us.
Comments for the Authors:
1- The Simple Summary and Abstract state 305 herds; the Results also mention 305 enrolled; however, the Herd Selection section in Methods indicates that 304 herds “could be examined.” Please clarify whether 304 was the intended sample or the number actually examined, and make this consistent throughout the manuscript.
- AU: We thank the reviewer for pointing out the discrepancy. As described in the Materials and Methods, 304 herds were originally planned for examination. Due to delayed responses from some farmers, one additional herd was included, resulting in a total of 305 herds. This is clarified in the Results section. (p.8, L298-300)
2- The “Laboratory analysis” section describes several criteria and that should be supported with references to standard protocols or relevant previous studies.
- AU: We have revised the "Laboratory analysis" section and included references to relevant studies where applicable. In general, all laboratory procedures were conducted according to the guidelines of the German Veterinary Medical Society (DVG). These guidelines are widely recognized as the national standard in Germany. Where additional or deviating procedures were applied, we have now clearly indicated this and provided appropriate references. (p. 5, L191,202,205,207)
3- Please define AMS in the Table 2 in a footnote.
- AU: We have corrected the terminology in Tables 2 and 8 by replacing "AMS" with "robotic milking system" to ensure consistency throughout the manuscript. (p. 6-7, Table 1 and p.19 ,Table 8)
The manuscript would benefit from careful proofreading. For example:
Line 430: “Since that is numerically lower than our results and it also may indicate a slight upward trend” is incomplete, as “Since” introduces a dependent clause without a corresponding main clause. Consider rephrasing for clarity.
- AU: Done. Plus, the manuscript underwent a professional proofread. (p. 21, L467-469)
Table 2: Correct “Teat clealiness Score” to “Teat cleanliness Score” and “Deep beeded Cubicles” to “Deep bedded cubicles.”
- AU: Corrected (p.9, Table 2)
The sentence across lines 447–449 contains a long sequence of pathogen names, prevalence values, and rankings, which makes it heavy to read. Consider rephrasing to improve flow and readability.
- AU: We have adjusted the relevant part. (p.21, L495-499)
Reviewer 2 Report
Comments and Suggestions for Authors
The paper, titled "Risk factors for intramammary infections on Bavarian dairy farms - a herd level analysis," addresses an important and timely topic for the dairy industry. I found the subject matter of the article quite fascinating and, honestly, I read the manuscript with great interest. The paper defintely aligns well with the scope of the journal, no question there.
Main Question Addressed
core question this research tackles is to identify the apparent prevalence of mastitis pathogens and, more importantly, to pinpoint herd-level risk factors associated with intramammary infections (IMI) in dairy herds located in Bavaria, Germany. The also look at specific pathogens like Strep. uberis, Strep. dysgalactiae, Strep. agalactiae, S. aureus, E. coli, and non-aureus staphylococci (NAS).
Originality and Relevance to the Field
Oh, I definitely consider this topic both original and highly relevant to the field of veterinary medicine and dairy science. Mastitis a huge economic burden and animal welfare concern globally, so any study that helps us understand and prevent it is valuable.
It specificaly addresses a gap by providing a re-evaluation of risk factors for IMI in the Bavarian dairy region, taking into account recent legislative changes concerning antibiotic use and the evolving structure of dairy farms (fewer, larger farms, and more robotic milking systems). Previos regional studies were somewhat older or couldn't assess seasonal factors, so this is a crucial update. Plus it focuses on the predominant Simmental breed in Bavaria, which is a neat, breed-specific angle.
Contribution to the Subject Area
This paper adds quite a bit compared to other published material. First off, it provides a current, unbiased, and comprehensive snapshot of mastitis pathogen prevalence in Bavarian dairy herds, systematically sampling all lactating cows from randomly selected farms. That's a strong methodological point.
It confirms some known risk factors, like the importance of bedding and dry-off treatment, but also identifies newer or re-emphasized ones, such as the increased prevalence of NAS with automated milking systems and the unexpected association of clean milking systems with higher S. aureus prevalence (which they acknowledge needs more investigation, obviously). The finding regarding Strep. agalactiae and water trough cleaning, suggesting an environmental component, is also really interesting and kind of challenges the old textbooks on contagious pathogens. They've given us a solid baseline for current regional challenges.
However, I believe that in its current form, it has several shortcomings that need addressing.
Specific comments
Introduction
The introduction sets the stage pretty well, outlining the importance of mastitis and the background on pathogen classification. However, I think it could be even stronger by perhaps briefly highlighting the unique aspects of Bavarian dairy farming earlier – the Simmental breed dominance, for instance (cite: 10.3389/fvets.2023.1141286) – to immediately establish why a regional study is so vital and not just, well, another study. It'd draw the reader in quicker. Also, the phrase about "legislative changes regarding antibiotic usage" is mentioned, which is good, but a tiny bit more context on what those changes entail for Bavarian farmers specifically might be helpful for an international audience.
page 2, line 43, the phrase "the impact on animal welfare is high" is a bit general; maybe a stronger verb or a slightly more specific adjective could make it more impactful, like "profound" or "significant." Also double-check for any slight redundancies in phrasing; I noticed a couple of times where a point was reiterated very closely.
Materials and Methods
Regardng the methodology, a key improvement would be to elaborate more on the rationale for excluding farms with less than 200 kg daily shipped milk. While assumption of herd size less than 10 cows is noted, it would be beneficial to explain why these smaller herds were deemed unsuitable for the study or how their exclusion might affect the generalizability of results, particularly given Bavaria's smaller average herd size. It just seems like a lot of farms were dropped, you know? Additionally, in the laboratory analysis section, for cases where "no species could be determined after repeated microbiological examination and MALDI-TOF," classifying them simply as "esculin-positive and esculin-negative streptococci" seems a bit broad. Could the authors consider discussing the potential implications of these unidentifiable isolates on the prevalence data? Also, a minor point, but the "July 2023 and July 2024" dates for farm visits in the abstract are a little confusing compared to the "2014 and 2023" mentioned later in relation to other studies; perhaps a quick clarification of the study's actual data collection period would be good. The statistical analysis section is quite detailed, which is good, but some of the predictor descriptions in Table 1 could be clearer. For instance, what exactly constitutes "other" for milking systems or dip agent category? More specific examples would be useful.
In the "Herd selection" section, when you talk about the randomization in Microsoft Excel®, maybe just mentioning the specific function or method used (like RANDBETWEEN or similar) would be a nice, small detail for transparency. as the description of scores (hygiene, hock, teat), ensure the score ranges are consistently presented (e.g., always "1 to 4" or "1-4" rather than mixing them up). Also on page 4, lines 165-169, the detailed explanation of bedding types within parentheses is a bit of a mouthful; perhaps it could be rephrased or presented as a small footnote if it clutters the main text.
Results
results presented clearly, and the tables are well-structured. However, when discussing the prevalence of mastitis pathogens, particularly for NAS, the percentage in QMS (5%) and then "in 63% of all healthy quarters" can be a tad confusing without a clearer explanation of what "healthy quarters" means in the context of NAS presence. It implies that a pathogen can be present without causing an elevated SCC, which is an important nuance. Also, for Strep. agalactiae and E. coli, which had very low prevalences, the authors state the outcome was binary (herd infected: yes/no). While understandable, it might be beneficial to provide the absolute number of positive QMS or cows for these pathogens alongside the herd-level presence, just for completeness, you know? It'd give a fuller picture.
In the "Prevalence of mastitis pathogens" section, when describing the percentage of contaminated samples (1.5% with >2 pathogen types), clarify if that 1.5% is of the total QMS samples or of only the contaminated ones for absolute clarity. On page 8, line 282, "The best 25% of farms had less than 13% of cows with a hygiene score ≥3 and 0% of cows had a hock score ≥2," this sentence is a tad long; perhaps breaking it into two shorter sentences would improve flow. Also, ensure consistent capitalization of "Table" when referring to tables throughout the text.
Discussion
The discussion is generally interprets the findings well and compares them to previous research. However there are a few instances where the conclusions could be more consistently aligned with the evidence presented. as example, the association of "clean milking system hygiene" with a higher within-herd prevalence of S. aureus is highlighted as counterintuitive. While the authors rightly state that "no cause effect or time inference is possible" due to the cross-sectional design, a bit more speculation (with appropriate caveats) on why this association might exist (e.g., perhaps farms with higher S. aureus problems prior to the study became hyper-vigilant about hygiene, but the existing infections were already established) could add depth. Otherwise, it just kinda hangs there. Similarly the finding that "absence of teat cleaning was associated with a lower Strep. dysgalactiae within-herd-prevalence" is interesting, but the proposed explanation ("well maintained cubicles resulted in better cow and udder hygiene") feels a little speculative without more direct supporting data within this study. Its a plausible theory, but it's not directly evidenced. The point about Brown Swiss cattle being more susceptible to Strep. uberis is also intriguing, especially given their supposed resistance to IMI. More discusion on potential breed-specific management practices or physiological differences that might explain this observed susceptibility would be great, even if it's just a hypothesis for future research.
page 22, line 468, "Groh et al. (2023) was previously unable to find an association..." I noticed that the year in the reference list is 2022 for that specific paper. Just a small typo in the year there, easy fix. Some of the sentences, particularly when comparing findings to other studies, tend to be quite long. Shortening them or breaking them up might help improve readability. I also spotted a couple of missing commas here and there, mainly in longer sentences, which can affect clarity.
You know, after going through the whole paper, it's really quite thorough and important for understanding mastitis in Bavaria. That said, I had a thought – it'd be super neat if the authors could add a short section on future perspectives, especially looking at how precision livestock farming (PLF) technologies could play a role here.
Given that they've already touched on things like automated milking systems and even observed cow behavior (agitated cows during milking!), it feels like a natural next step for the discussion. Imagine how these technologies could help with mastitis!
For example, they could talk about:
Real-time monitoring: How wearable sensors or in-parlor systems could constantly track individual cow health parameters, like tiny changes in activity, rumination patterns, or even very early shifts in milk conductivity or somatic cell count. This could catch mastitis way earlier, perhaps even before it's subclinical.
I suggest citing 10.3390/ani15030458 and 10.3389/fanim.2025.1547395.
Targeted intervention: If you have that real-time data, farmers could intervene much more precisely, maybe even reducing the need for broad-spectrum antibiotics which ties into the legislative changes they mentioned. Its all about getting the right treatment to the right cow at the right time.
Data-driven risk prediction: The study identifies a bunch of herd-level risk factors. PLF tools could help integrate all that environmental, management, and individual cow data to build even more sophisticated predictive models, maybe specific to Bavarian farms or even Simmental cows.
Automation in hygiene: They talk about milking hygiene and bedding. Could PLF bring in more automated or data-driven ways to manage these aspects, further reducing pathogen exposure?
It just seems like a really important and evolving area that directly relates to their findings on prevalence and risk factors. It would show the paper isn't just a snapshot but also looks forward to how these insights can actually be applied in the real world of modern dairy farming. Plus, it would make the paper feel even more, umm, forward-thinking, ya know?
It would beneficial for the authors to consider adding a short section on how to share these findings with the public, especially through social media. In our field of animal science, there's just so much misinformation, and getting accurate research out there is, like, super important. Discusing how this work on mastitis risk factors in Bavarian dairy farms could reach beyond just academics—maybe on Twitter or even Instagram—would be really useful. It's about being transparent and helping folks understand complex topics better.
In parallel vein, a study focused on utilizing Instagram illustrates how social media can serve as an effective tool (10.3168/jds.2024-25347). This study underscores the power of social media in conveying complex topics, such as the prevalence of intramammary infections and their associated herd-level risk factors, to a broad audience. Such initiatives complement the role of influencers in promoting evidence based animal health communication by providing tangible examples of how digital platforms can foster community engagement and awareness in specialized areas.
Conclusions
are consistent with the evidence and arguments, summarizing the key findings and addressing the main question posed by the research quite well. They efectively underscore the importance of adequate bedding, dry-off treatment, and milking hygiene. That statement that the study "provides an unbiased view of the presence of relevant mastitis pathogens on small- to mid-sized farms" is a fair and important claim given the methodology. Overall they've done a good job wrapping things up.
The conclusions are concise and effective. Just a quick read-through for any minor grammatical slips would be useful. Nothing major jumps out, but a fresh pair of eyes could catch tiny things.
References
The references seem generally appropriate and cover a wide range of relevant literature, both recent and foundational. I noticed a good mix of journal articles and some important organizational guidelines (IDF, NMC). There doesn't appear to be any obvious omissions or irrelevant citations. Everything seems to be well-supported.
Everithing here looks solid. Just a general suggestion to double-check the formatting against the journal's specific guidelines one last time; sometimes journals have really particular preferences for things like author initials, journal abbreviations, o DOI presentation.
Here's a little list of spots where I think references could be added, or are needed:
On page 2, line 44, talking about "Affected animals can show clear behavioral changes, such as shorter lying times". While there's a citation (reference 3), more recent or perhaps other common behavioral indicators could be added, and cited, if they're not already covered. It's a broad statement, so another paper or two reinforcing that idea wouldn't hurt.
Line 46: regarding rumination time and feeding behavior, I suggest citing: 10.3168/jds.2025-26385.
Still on page 2, around lines 59-61, where it discusses how "some pathogens, such as Strep. dysgalactiae [14] and the heterogeneous group of non-aureus staphylococci (NAS), can exhibit both environmental and contagious transmission characteristics [15,16]." While they have references, this is a pretty key point about pathogen behavior, so perhaps a more recent review or a seminal paper supporting the Strep. dysgalactiae dual nature might be beneficial. Just to really hammer it home.
On page 2, lines 77-78, "Cows with a higher parity are more prone to mastitis [22,23]. Similarly, high milk yield has been associated with an increased risk of mastitis [24]." These are foundational concepts, but sometimes adding a very recent, large-scale meta-analysis, if one exists, could provide even stronger support. Or perhaps a very classic, widely-cited paper if 22-24 are more recent.
When they introduce "Known risk factors for IMI include for instance age and milk yield" on page 2, line 77, it might be worth adding a general review paper on mastitis risk factors right at the beginning of that paragraph, setting the stage before diving into specifics.
On page 3, lines 109-110, where it states "the dairy industry has developed towards fewer but larger dairy farms and the number of farms with robotic milking is steadily increasing." This is a generally known trend, but if there's a specific agricultural statistics report or an economic analysis paper for Germany or Bavaria that confirms these trends, adding that reference would make the statement even more robust. It's a factual claim about industry change.
In the Discussion, page 21, lines 437-438, "The higher milking frequency of automatic milking systems compared to the traditional 2x/d milking might also affect their presence in the teat canal." This is a bit of a logical inference. If there are any studies that specifically link increased milking frequency (from AMS) to teat canal health or NAS presence, that would be a very strong citation to add there. It's not just that AMS exist, but how they influence things.
Also, on page 22, lines 471-475, they talk about organic farmers refraining from antibiotic treatment to avoid economic strain or losing organic certification. While this is a plausible explanation, if there are studies or guidelines from organic farming bodies that explicitly detail these specific economic or certification pressures related to mastitis treatment, those would be excellent references to include. It lends weight to their explanation.
When discussing the seasonality of mastitis on page 24, lines 553-556, they mention that "Bechtold et al. (2024) found more environmental pathogens in quarter milk samples during the summer months in the same region and cows shed Strep. uberis more during the hot season [25]." If there are other studies that have investigated seasonal effects on herd-level prevalence specifically (even if they found no effect, like this paper), citing them could reinforce the discussion point.
Tables and Figures
The tables are mostly well-organized and easy to understand. Table 1 (Evaluated risk factors) is good for showing what was assessed. However, some of the sub-bullets could be re-formatted a little cleaner or with more consistent punctuation, just to make it easier on the eyes. In Table 2 (Herd description and farm analysis), the use of medians and percentiles is appropriate for the data, but maybe adding the sample size (n) for each percentage might be useful, particularly for the smaller "Other" categories within breeds or housing, to show the base. Tables 5 to 10 (multivariate models) are clear, presenting prevalence ratios/odds ratios, confidence intervals, and p-values, which is exactly what one expects. Everything looks presentable.
For Table 1, some of the descriptions under "Farm- and herd structure" and "Milking and milking system" are quite detailed. While useful, sometimes the parenthetical explanations (like (biocontrol of purchased animals¹) or (never, automatic, manual by pushing the claw down)) could perhaps be integrated more smoothly into the main variable description or, again, considered for a more detailed footnote if they make the table too dense. For all tables, ensure the alignment of numbers and decimals is consistent throughout for optimal visual presentation.
Author Response
Thank you for carefully reviewing our manuscript and providing constructive and insightful feedback. We greatly appreciate the time and effort you dedicated to evaluating our work. To enhance clarity and readability, MDPI Author Services professionally proofread the manuscript, and we reformatted all tables for consistency and visual clarity. All changes made in response to your comments are highlighted in yellow in the revised manuscript. Our point-by-point responses to your comments are provided below.
The paper, titled "Risk factors for intramammary infections on Bavarian dairy farms - a herd level analysis," addresses an important and timely topic for the dairy industry. I found the subject matter of the article quite fascinating and, honestly, I read the manuscript with great interest. The paper defintely aligns well with the scope of the journal, no question there.
- AU: We would like to thank you sincerely for your positive and encouraging feedback on our manuscript. We are very grateful that you found the topic relevant and interesting, and we are glad to hear that the paper is well within the scope of the journal. Your kind words are a great source of motivation for us, and we are very grateful for the time and effort you have taken to review our work.
Main Question Addressed
core question this research tackles is to identify the apparent prevalence of mastitis pathogens and, more importantly, to pinpoint herd-level risk factors associated with intramammary infections (IMI) in dairy herds located in Bavaria, Germany. The also look at specific pathogens like Strep. uberis, Strep. dysgalactiae, Strep. agalactiae, S. aureus, E. coli, and non-aureus staphylococci (NAS).
Originality and Relevance to the Field
Oh, I definitely consider this topic both original and highly relevant to the field of veterinary medicine and dairy science. Mastitis a huge economic burden and animal welfare concern globally, so any study that helps us understand and prevent it is valuable.
It specificaly addresses a gap by providing a re-evaluation of risk factors for IMI in the Bavarian dairy region, taking into account recent legislative changes concerning antibiotic use and the evolving structure of dairy farms (fewer, larger farms, and more robotic milking systems). Previos regional studies were somewhat older or couldn't assess seasonal factors, so this is a crucial update. Plus it focuses on the predominant Simmental breed in Bavaria, which is a neat, breed-specific angle.
- AU: We would like to thank you sincerely for your thoughtful and detailed feedback on our manuscript. We are pleased that you consider the topic to be both original and highly relevant to veterinary medicine and dairy science. We appreciate your recognition of the study’s contribution to addressing the economic and animal welfare implications of mastitis and the necessity of reevaluating risk factors in light of recent legislative changes and evolving farm structures. Your positive remarks on the seasonal aspects and focus on the Simmental breed are especially encouraging. We are truly grateful for your kind words and for the time you invested in carefully reviewing our work.
Contribution to the Subject Area
This paper adds quite a bit compared to other published material. First off, it provides a current, unbiased, and comprehensive snapshot of mastitis pathogen prevalence in Bavarian dairy herds, systematically sampling all lactating cows from randomly selected farms. That's a strong methodological point.
It confirms some known risk factors, like the importance of bedding and dry-off treatment, but also identifies newer or re-emphasized ones, such as the increased prevalence of NAS with automated milking systems and the unexpected association of clean milking systems with higher S. aureus prevalence (which they acknowledge needs more investigation, obviously). The finding regarding Strep. agalactiae and water trough cleaning, suggesting an environmental component, is also really interesting and kind of challenges the old textbooks on contagious pathogens. They've given us a solid baseline for current regional challenges.
However, I believe that in its current form, it has several shortcomings that need addressing.
Specific comments
Introduction
The introduction sets the stage pretty well, outlining the importance of mastitis and the background on pathogen classification. However, I think it could be even stronger by perhaps briefly highlighting the unique aspects of Bavarian dairy farming earlier – the Simmental breed dominance, for instance (cite: 10.3389/fvets.2023.1141286) – to immediately establish why a regional study is so vital and not just, well, another study. It'd draw the reader in quicker.
- AU: Thank you very much. We have moved the section on Simmental cows and the Bavarian region further up and also added the suggested reference. (p. 2, L52-59,52)
Also, the phrase about "legislative changes regarding antibiotic usage" is mentioned, which is good, but a tiny bit more context on what those changes entail for Bavarian farmers specifically might be helpful for an international audience.
- AU: Done. we have specified the recent legislative changes. (p.3, L112-118)
page 2, line 43, the phrase "the impact on animal welfare is high" is a bit general; maybe a stronger verb or a slightly more specific adjective could make it more impactful, like "profound" or "significant." Also double-check for any slight redundancies in phrasing; I noticed a couple of times where a point was reiterated very closely.
- AU: Done. (p.2, L46)
Materials and Methods
Regardng the methodology, a key improvement would be to elaborate more on the rationale for excluding farms with less than 200 kg daily shipped milk. While assumption of herd size less than 10 cows is noted, it would be beneficial to explain why these smaller herds were deemed unsuitable for the study or how their exclusion might affect the generalizability of results, particularly given Bavaria's smaller average herd size. It just seems like a lot of farms were dropped, you know?
- AU: Thank you for your comment. We agree that the exclusion of farms with a daily milk yield of less than 200kg warrants clarification. We have now added this explanation to the material and methods section (L136-138) and further discuss in the discussion section that our results should be extrapolated only to herds with more than 10 and up to 250 cows. (p.21, L442-443)
Additionally, in the laboratory analysis section, for cases where "no species could be determined after repeated microbiological examination and MALDI-TOF," classifying them simply as "esculin-positive and esculin-negative streptococci" seems a bit broad. Could the authors consider discussing the potential implications of these unidentifiable isolates on the prevalence data?
- AU: Thank you for this comment. Indeed, only a very small proportion of isolates (< 0.1% of all QMS) could not be definitively identified beyond esculin-positive or -negative streptococci – even with the MALDI-TOF. While these unidentifiable isolates are unlikely to substantially affect the reported prevalence of specific pathogens, we have added a brief note in the discussion section to acknowledge this point. (p.22, L499-502)
Also, a minor point, but the "July 2023 and July 2024" dates for farm visits in the abstract are a little confusing compared to the "2014 and 2023" mentioned later in relation to other studies; perhaps a quick clarification of the study's actual data collection period would be good.
- AU: Done, we have added January 2014 to December 2023 for the study by Bechtold et al. (p.3, L108-109)
The statistical analysis section is quite detailed, which is good, but some of the predictor descriptions in Table 1 could be clearer. For instance, what exactly constitutes "other" for milking systems or dip agent category? More specific examples would be useful.
- AU: Done, we have provided explanations for “others” in the footnotes. (p.6-7, Table 1)
In the "Herd selection" section, when you talk about the randomization in Microsoft Excel®, maybe just mentioning the specific function or method used (like RANDBETWEEN or similar) would be a nice, small detail for transparency.
- AU: Done, we have added to the materials and methods section that we used the RAND() function. (p.4, L150)
As the description of scores (hygiene, hock, teat), ensure the score ranges are consistently presented (e.g., always "1 to 4" or "1-4" rather than mixing them up).
- AU: We couldn't find any inconsistencies in the description of the scores, but thank you for pointing this out. We have revised it again and agree with you, of course.
Also, on page 4, lines 165-169, the detailed explanation of bedding types within parentheses is a bit of a mouthful; perhaps it could be rephrased or presented as a small footnote if it clutters the main text.
- Thank you for pointing that out. We have removed the section and optimised it as a footnote in Tables 1 and 5, so that the materials and methods section reads more easily. (p. 6-7, Table 1; p.8, Table 5)
Results
results presented clearly, and the tables are well-structured.
AU: Thank you.
However, when discussing the prevalence of mastitis pathogens, particularly for NAS, the percentage in QMS (5%) and then "in 63% of all healthy quarters" can be a tad confusing without a clearer explanation of what "healthy quarters" means in the context of NAS presence. It implies that a pathogen can be present without causing an elevated SCC, which is an important nuance.
- AU: Thank you for pointing that out. We have now clarified in the text that we are referring to pathogen-positive QMS. (p.11, L348-351)
Also, for Strep. agalactiae and E. coli, which had very low prevalences, the authors state the outcome was binary (herd infected: yes/no). While understandable, it might be beneficial to provide the absolute number of positive QMS or cows for these pathogens alongside the herd-level presence, just for completeness, you know? It'd give a fuller picture.
- AU: Done, we have added the relevant information to the results section. (p.11, L345-347)
In the "Prevalence of mastitis pathogens" section, when describing the percentage of contaminated samples (1.5% with >2 pathogen types), clarify if that 1.5% is of the total QMS samples or of only the contaminated ones for absolute clarity.
- AU: Done. (p. 11, L331-333)
On page 8, line 282, "The best 25% of farms had less than 13% of cows with a hygiene score ≥3 and 0% of cows had a hock score ≥2," this sentence is a tad long; perhaps breaking it into two shorter sentences would improve flow.
- AU: Done. (p. 8, L306)
Also, ensure consistent capitalization of "Table" when referring to tables throughout the text.
- AU: Done, we have thoroughly checked the manuscript for consistent capitalization.
Discussion
The discussion is generally interprets the findings well and compares them to previous research. However there are a few instances where the conclusions could be more consistently aligned with the evidence presented.
As example, the association of "clean milking system hygiene" with a higher within-herd prevalence of S. aureus is highlighted as counterintuitive. While the authors rightly state that "no cause effect or time inference is possible" due to the cross-sectional design, a bit more speculation (with appropriate caveats) on why this association might exist (e.g., perhaps farms with higher S. aureus problems prior to the study became hyper-vigilant about hygiene, but the existing infections were already established) could add depth. Otherwise, it just kinda hangs there.
- AU: Thank you for this valuable comment. We agree that the observed association between "clean milking system hygiene" and higher within-herd prevalence of S. aureus may appear counterintuitive at first glance. As suggested, we have revised the manuscript to include a brief, cautious discussion of potential explanations. Specifically, we now speculate that farms with a history of S. aureus problems may have adopted stricter hygiene protocols prior to the study, but existing infections may have persisted, thereby explaining the observed association. We have emphasized that, due to the cross-sectional nature of the study, no causal inferences can be drawn. The relevant section has been updated accordingly to provide more context and interpretation for readers. (p. 23, L567-570)
Similarly the finding that "absence of teat cleaning was associated with a lower Strep. dysgalactiae within-herd-prevalence" is interesting, but the proposed explanation ("well maintained cubicles resulted in better cow and udder hygiene") feels a little speculative without more direct supporting data within this study. Its a plausible theory, but it's not directly evidenced.
- AU: We acknowledge that the proposed explanation for the association between the absence of teat cleaning and lower Streptococcus dysgalactiae within-herd prevalence was somewhat speculative. In response, we have revised the manuscript to clarify that this is a hypothesis rather than a conclusion, and we now explicitly state that no direct evidence supporting this mechanism was obtained in our study. The text has been adjusted to reflect this limitation while retaining the possible interpretation as a basis for future research. (p. 23, L558-560)
The point about Brown Swiss cattle being more susceptible to Strep. uberis is also intriguing, especially given their supposed resistance to IMI. More discusion on potential breed-specific management practices or physiological differences that might explain this observed susceptibility would be great, even if it's just a hypothesis for future research.
- AU: Thank you for highlighting this important point. We agree that the finding regarding increased susceptibility of Brown Swiss cattle to Streptococcus uberis is intriguing, especially considering their commonly perceived resistance to intramammary infections. In response, we have expanded the discussion to include potential explanations. While our current data do not allow for definitive conclusions, we have framed these points as hypotheses to be explored in future studies. (p.24, L578-587)
page 22, line 468, "Groh et al. (2023) was previously unable to find an association..." I noticed that the year in the reference list is 2022 for that specific paper. Just a small typo in the year there, easy fix.
- AU: Done. (L532,614,110)
Some of the sentences, particularly when comparing findings to other studies, tend to be quite long. Shortening them or breaking them up might help improve readability. I also spotted a couple of missing commas here and there, mainly in longer sentences, which can affect clarity.
- AU: Thank you very much for the helpful tip, we have had it proofread.
You know, after going through the whole paper, it's really quite thorough and important for understanding mastitis in Bavaria. That said, I had a thought – it'd be super neat if the authors could add a short section on future perspectives, especially looking at how precision livestock farming (PLF) technologies could play a role here.
Given that they've already touched on things like automated milking systems and even observed cow behavior (agitated cows during milking!), it feels like a natural next step for the discussion. Imagine how these technologies could help with mastitis!
- AU: Thank you very much for your constructive feedback and valuable suggestion to include a perspective on future developments, particularly with regard to PLF technologies. We agree that PLF technologies have great potential for preventing and managing mastitis.
However, as our study was based on one-time visits to farms, our data only allow us to identify associations rather than causal relationships. Therefore, we believe that it would be difficult to speculate on future developments based on our current findings. We did, however, make a brief note regarding robotic milking systems, as these have been identified as a risk factor for mastitis on dairy farms (p.22, L477-485). Apart from this, we do not discuss precision livestock farming further in this paper, as this would take us beyond the scope of our current investigation.
We greatly appreciate your suggestion and recognise the importance of this topic for future research, especially in the context of a long-term study.
Thanks again for your insightful thoughts!
For example, they could talk about:
Real-time monitoring: How wearable sensors or in-parlor systems could constantly track individual cow health parameters, like tiny changes in activity, rumination patterns, or even very early shifts in milk conductivity or somatic cell count. This could catch mastitis way earlier, perhaps even before it's subclinical.
I suggest citing 10.3390/ani15030458 and 10.3389/fanim.2025.1547395.
Targeted intervention: If you have that real-time data, farmers could intervene much more precisely, maybe even reducing the need for broad-spectrum antibiotics which ties into the legislative changes they mentioned. Its all about getting the right treatment to the right cow at the right time.
Data-driven risk prediction: The study identifies a bunch of herd-level risk factors. PLF tools could help integrate all that environmental, management, and individual cow data to build even more sophisticated predictive models, maybe specific to Bavarian farms or even Simmental cows.
Automation in hygiene: They talk about milking hygiene and bedding. Could PLF bring in more automated or data-driven ways to manage these aspects, further reducing pathogen exposure?
It just seems like a really important and evolving area that directly relates to their findings on prevalence and risk factors. It would show the paper isn't just a snapshot but also looks forward to how these insights can actually be applied in the real world of modern dairy farming. Plus, it would make the paper feel even more, umm, forward-thinking, ya know?
It would beneficial for the authors to consider adding a short section on how to share these findings with the public, especially through social media. In our field of animal science, there's just so much misinformation, and getting accurate research out there is, like, super important. Discusing how this work on mastitis risk factors in Bavarian dairy farms could reach beyond just academics—maybe on Twitter or even Instagram—would be really useful. It's about being transparent and helping folks understand complex topics better.
In parallel vein, a study focused on utilizing Instagram illustrates how social media can serve as an effective tool (10.3168/jds.2024-25347). This study underscores the power of social media in conveying complex topics, such as the prevalence of intramammary infections and their associated herd-level risk factors, to a broad audience. Such initiatives complement the role of influencers in promoting evidence based animal health communication by providing tangible examples of how digital platforms can foster community engagement and awareness in specialized areas.
- AU: We thank the reviewer for this thoughtful suggestion. We fully agree that PLF technologies represent an exciting and important future avenue for mastitis control. However, given the cross-sectional design and the specific scope of our dataset, a more detailed exploration of PLF applications would be highly speculative in the present context. To acknowledge this, we have added a brief statement in the discussion highlighting PLF as a potential area for future research. With regard to science communication and social media outreach, we very much share the reviewer’s view that choosing appropriate channels to disseminate scientific findings is of great importance. We see considerable potential here, particularly in reaching farmers directly. The results of this study are still being further evaluated, and it is certainly our goal to ensure that Bavarian dairy farmers (and farmers elsewhere) gain access to the relevant insights in a clear and practical way. We believe this is essential to maximize the impact of research in the field of animal health and mastitis prevention.
Conclusions
are consistent with the evidence and arguments, summarizing the key findings and addressing the main question posed by the research quite well. They efectively underscore the importance of adequate bedding, dry-off treatment, and milking hygiene. That statement that the study "provides an unbiased view of the presence of relevant mastitis pathogens on small- to mid-sized farms" is a fair and important claim given the methodology. Overall they've done a good job wrapping things up.
The conclusions are concise and effective.
- AU: Thank you.
Just a quick read-through for any minor grammatical slips would be useful. Nothing major jumps out, but a fresh pair of eyes could catch tiny things.
- AU: Thank you very much for the helpful tip, we have had it proofread.
References
The references seem generally appropriate and cover a wide range of relevant literature, both recent and foundational. I noticed a good mix of journal articles and some important organizational guidelines (IDF, NMC). There doesn't appear to be any obvious omissions or irrelevant citations. Everything seems to be well-supported.
Everithing here looks solid. Just a general suggestion to double-check the formatting against the journal's specific guidelines one last time; sometimes journals have really particular preferences for things like author initials, journal abbreviations, o DOI presentation.
- AU: Thank you for your helpful suggestion. We have carefully double-checked the formatting of all references against the journal’s specific guidelines, including author initials, journal abbreviations, and DOI presentation. All references have now been reviewed and adjusted where necessary to ensure full compliance with the required style.
Here's a little list of spots where I think references could be added, or are needed:
On page 2, line 44, talking about "Affected animals can show clear behavioral changes, such as shorter lying times". While there's a citation (reference 3), more recent or perhaps other common behavioral indicators could be added, and cited, if they're not already covered. It's a broad statement, so another paper or two reinforcing that idea wouldn't hurt.
- AU: Done. We have actually added two additional sources that also address the behavioral changes in affected animals and provide an additional perspective on lying behavior. (p. 2, L47)
Line 46: regarding rumination time and feeding behavior, I suggest citing: 10.3168/jds.2025-26385.
- AU: We thank the reviewer for the suggestion. However, after careful consideration, we have decided not to include the suggested references as they do not directly address rumination time and feeding behaviour in connection with mastitis.
Still on page 2, around lines 59-61, where it discusses how "some pathogens, such as Strep. dysgalactiae [14] and the heterogeneous group of non-aureus staphylococci (NAS), can exhibit both environmental and contagious transmission characteristics [15,16]." While they have references, this is a pretty key point about pathogen behavior, so perhaps a more recent review or a seminal paper supporting the Strep. dysgalactiae dual nature might be beneficial. Just to really hammer it home.
- AU: Done. We have added further sources that describe the specific characteristics of Strep. dysgalactiae. (p.2, L75)
On page 2, lines 77-78, "Cows with a higher parity are more prone to mastitis [22,23]. Similarly, high milk yield has been associated with an increased risk of mastitis [24]." These are foundational concepts, but sometimes adding a very recent, large-scale meta-analysis, if one exists, could provide even stronger support. Or perhaps a very classic, widely-cited paper if 22-24 are more recent.
- AU: Done. (p. 3, L93,94)
When they introduce "Known risk factors for IMI include for instance age and milk yield" on page 2, line 77, it might be worth adding a general review paper on mastitis risk factors right at the beginning of that paragraph, setting the stage before diving into specifics.
- AU: Done. (p.3, L92)
On page 3, lines 109-110, where it states "the dairy industry has developed towards fewer but larger dairy farms and the number of farms with robotic milking is steadily increasing." This is a generally known trend, but if there's a specific agricultural statistics report or an economic analysis paper for Germany or Bavaria that confirms these trends, adding that reference would make the statement even more robust. It's a factual claim about industry change.
- AU: Done. (p. 2, L120)
In the Discussion, page 21, lines 437-438, "The higher milking frequency of automatic milking systems compared to the traditional 2x/d milking might also affect their presence in the teat canal." This is a bit of a logical inference. If there are any studies that specifically link increased milking frequency (from AMS) to teat canal health or NAS presence, that would be a very strong citation to add there. It's not just that AMS exist, but how they influence things.
- AU: Done. (p.22, L476,477)
Also, on page 22, lines 471-475, they talk about organic farmers refraining from antibiotic treatment to avoid economic strain or losing organic certification. While this is a plausible explanation, if there are studies or guidelines from organic farming bodies that explicitly detail these specific economic or certification pressures related to mastitis treatment, those would be excellent references to include. It lends weight to their explanation.
- AU: Thank you for your valuable feedback. We have added a brief note to the discussion section to clarify the circumstances in which an animal loses its organic status due to antibiotic treatment. Specifically, we have indicated that, in the EU, an animal loses its organic status if it is treated with antibiotics more than three times per year. In the US, however, any antibiotic treatment leads to the loss of organic certification. We have retained the original references (European Commission regulations and USDA guidelines) as these provide the necessary context for the standards governing organic production. We believe that this additional clarification improves the explanation without the need for further references. (p. 23, L527-529)
When discussing the seasonality of mastitis on page 24, lines 553-556, they mention that "Bechtold et al. (2024) found more environmental pathogens in quarter milk samples during the summer months in the same region and cows shed Strep. uberis more during the hot season [25]." If there are other studies that have investigated seasonal effects on herd-level prevalence specifically (even if they found no effect, like this paper), citing them could reinforce the discussion point.
- AU: We agree that including studies on the seasonal effects on herd-level mastitis prevalence would strengthen the discussion. Despite our efforts, however, we were unable to find studies that specifically address seasonal variation in herd-level mastitis prevalence. It would also have been valuable for our discussion to have such sources for comparison.
Tables and Figures
The tables are mostly well-organized and easy to understand.
- AU: Thank you.
Table 1 (Evaluated risk factors) is good for showing what was assessed. However, some of the sub-bullets could be re-formatted a little cleaner or with more consistent punctuation, just to make it easier on the eyes.
- AU: Thank you very much for your valuable feedback regarding Table 1. We agree that the formatting and presentation of sub-bullets and descriptions could benefit from improved consistency and clarity. Given the large number of variables assessed, it has been challenging to balance detail and readability. We have revised the formatting of Table 1 to ensure more consistent punctuation and cleaner structure. (p.6-7, Table 1)
In Table 2 (Herd description and farm analysis), the use of medians and percentiles is appropriate for the data, but maybe adding the sample size (n) for each percentage might be useful, particularly for the smaller "Other" categories within breeds or housing, to show the base.
- AU: Thank you for your suggestion to include sample sizes (n) for each percentage in Table 2. To avoid overcrowding the table, which is already detailed, we have provided the specific sample sizes for the smaller 'Other' categories in the footnotes. However, since all the total sample sizes are listed at the top of the table, readers can also calculate the absolute values for each category if needed. We hope this solution strikes a balance between clarity and comprehensiveness. We are happy to provide further details if needed." (p.10, Foodnotes 2&7, Table2)
Tables 5 to 10 (multivariate models) are clear, presenting prevalence ratios/odds ratios, confidence intervals, and p-values, which is exactly what one expects. Everything looks presentable.
- AU: Thank you. We have nevertheless formatted it slightly for a better overview.
For Table 1, some of the descriptions under "Farm- and herd structure" and "Milking and milking system" are quite detailed. While useful, sometimes the parenthetical explanations (like (biocontrol of purchased animals¹) or (never, automatic, manual by pushing the claw down)) could perhaps be integrated more smoothly into the main variable description or, again, considered for a more detailed footnote if they make the table too dense. For all tables, ensure the alignment of numbers and decimals is consistent throughout for optimal visual presentation.
- AU: Thank you for this helpful suggestion. We fully understand the concern regarding the density of information in Table 1. Due to the large number of variables we collected, it has been challenging to present the data in a way that is both comprehensive and clearly structured. To address this, we have now added a more detailed explanation of the "Other" categories in the table footnotes to enhance transparency. Additionally, we have used italic font within parentheses to visually distinguish supplementary clarifications from the core variable descriptions, which we hope improves readability. We also reviewed all tables to ensure consistent alignment of numbers and decimal points for better visual presentation. (p.7, Foodnotes 5-9, Table1)
Reviewer 3 Report
Comments and Suggestions for Authors
This manuscript provides a highly relevant and timely analysis of mastitis pathogens and herd-level risk factors for intramammary infections (IMI) in Bavarian dairy farms. The dataset is exceptionally large (>57,000 quarter milk samples from ~14,700 cows, across 305 herds), which is a major strength and makes the findings particularly robust. The study is well structured, clearly written, and covers both microbiological and management aspects comprehensively.
Overall, the manuscript represents a valuable contribution to the field of dairy herd health. I have only a few minor points that should be addressed to improve clarity and strengthen interpretation.
Minor comments
Microbiological methodology – scope and limitations
The use of conventional culture methods (blood agar, esculin agar, Sabouraud agar, MALDI-TOF for selected isolates) is appropriate and aligns with current guidelines. However, the lack of molecular methods (PCR, sequencing, or 16S profiling) should be acknowledged as a limitation. Culture may underestimate or fail to detect fastidious or emerging pathogens (e.g., Mycoplasma spp., certain Corynebacteria). A short statement in the discussion would help balance the interpretation.
Pathogen identification – partial use of MALDI-TOF
The manuscript notes that not all colonies were identified by MALDI-TOF for logistical reasons. It would be useful to clarify briefly how colonies were prioritized for MALDI-TOF testing and to acknowledge the potential for rare species being underreported.
Statistical analysis – association vs. causation
The models applied (negative binomial and logistic regression) are appropriate. However, as this is a cross-sectional study, causality cannot be inferred. Some associations (e.g., “clean milking system associated with higher S. aureus prevalence”) may reflect reverse causality. A brief reminder in the conclusion that findings represent associations, not proven causal effects, would improve clarity.
Collinearity
The manuscript states that variables were checked for collinearity, but no details are provided. A short note in the methods (e.g., threshold values used, or examples of excluded variables) would enhance transparency. A full table is not required.
Presentation of prevalence data
At times, percentages are reported at different levels (quarter vs. cow vs. herd) without explicit distinction. For example, NAS prevalence is given in several places. Making it consistently clear whether results refer to quarters, cows, or herds will improve readability.
Typographical and stylistic issues
Line 91: “1,036,089 million dairy cows” → should be corrected to “1,036,089 dairy cows.”
Terms such as “clean” vs. “dirty” milking system could be replaced by more scientific phrasing (e.g., “high vs. low hygiene score”).
Interpretation of NAS
The group “non-aureus staphylococci (NAS)” is discussed as a whole. Given their heterogeneity in pathogenicity, it would be helpful to include one sentence in the discussion emphasizing that species-level differences exist, even if not all isolates were speciated.
Author Response
Thank you for carefully reviewing our manuscript and providing constructive and insightful feedback. We greatly appreciate the time and effort you dedicated to evaluating our work. To enhance clarity and readability, MDPI Author Services professionally proofread the manuscript, and we reformatted all tables for consistency and visual clarity. All changes made in response to your comments are highlighted in yellow in the revised manuscript. Our point-by-point responses to your comments are provided below.
This manuscript provides a highly relevant and timely analysis of mastitis pathogens and herd-level risk factors for intramammary infections (IMI) in Bavarian dairy farms. The dataset is exceptionally large (>57,000 quarter milk samples from ~14,700 cows, across 305 herds), which is a major strength and makes the findings particularly robust. The study is well structured, clearly written, and covers both microbiological and management aspects comprehensively.
Overall, the manuscript represents a valuable contribution to the field of dairy herd health. I have only a few minor points that should be addressed to improve clarity and strengthen interpretation.
- AU: We would like to sincerely thank you for your very positive and encouraging evaluation of our manuscript. We greatly appreciate your recognition of the size and strength of the dataset, as well as your comments on the structure and clarity of the paper. It is very motivating for us that you consider the study a valuable contribution to the field of dairy herd health. Thank you also for your constructive minor suggestions, which we are confident will help to further improve the clarity and interpretation of the manuscript.
Minor comments
Microbiological methodology – scope and limitations
The use of conventional culture methods (blood agar, esculin agar, Sabouraud agar, MALDI-TOF for selected isolates) is appropriate and aligns with current guidelines.
However, the lack of molecular methods (PCR, sequencing, or 16S profiling) should be acknowledged as a limitation.
Culture may underestimate or fail to detect fastidious or emerging pathogens (e.g., Mycoplasma spp., certain Corynebacteria). A short statement in the discussion would help balance the interpretation.
- AU: Done, we agree and have added a comment to the discussion. (p.21, L451-454)
Pathogen identification – partial use of MALDI-TOF
The manuscript notes that not all colonies were identified by MALDI-TOF for logistical reasons. It would be useful to clarify briefly how colonies were prioritized for MALDI-TOF testing and to acknowledge the potential for rare species being underreported.
- AU: Thank you for your comment. As noted in the "Materials and Methods" section, all NAS strains, questionable S. aureus isolates, and other pathogens that could not be clearly identified microbiologically, as well as all gram-negative pathogens, were prioritized for MALDI-TOF testing (p.5, L194,195,208,209,210,216,219). However, we acknowledge that MALDI-TOF may not always provide the highest accuracy for differentiating certain rare or atypical species, particularly in the case of streptococci, which could lead to potential misidentifications rather than underreporting. (p.21, L451-454)
Statistical analysis – association vs. causation
The models applied (negative binomial and logistic regression) are appropriate. However, as this is a cross-sectional study, causality cannot be inferred. Some associations (e.g., “clean milking system associated with higher S. aureus prevalence”) may reflect reverse causality. A brief reminder in the conclusion that findings represent associations, not proven causal effects, would improve clarity.
- AU: Done. We have added a statement about this in the discussion. (p.23/24, L570-573)
Collinearity
The manuscript states that variables were checked for collinearity, but no details are provided. A short note in the methods (e.g., threshold values used, or examples of excluded variables) would enhance transparency. A full table is not required.
- AU: Thank you very much for this helpful comment. We agree that additional clarification on the handling of collinearity improves transparency. In the revised version, we have added a short explanatory note to the materials and methods section (Statistical analysis). (p.6, L248-253)
Presentation of prevalence data
At times, percentages are reported at different levels (quarter vs. cow vs. herd) without explicit distinction. For example, NAS prevalence is given in several places. Making it consistently clear whether results refer to quarters, cows, or herds will improve readability.
- AU: Done, we have clarified at the respective points at which level the description is made. (L343,457,467,505,506,510,512,515,591)
Typographical and stylistic issues
Line 91: “1,036,089 million dairy cows” → should be corrected to “1,036,089 dairy cows.”
- AU: Done. (p.2, L54)
Terms such as “clean” vs. “dirty” milking system could be replaced by more scientific phrasing (e.g., “high vs. low hygiene score”).
- AU: Done (p.6-7, Table 1; p.17, Table 6)
Interpretation of NAS
The group “non-aureus staphylococci (NAS)” is discussed as a whole. Given their heterogeneity in pathogenicity, it would be helpful to include one sentence in the discussion emphasizing that species-level differences exist, even if not all isolates were speciated.
- AU: Done, we agree and have added a corresponding comment to the discussion. (p.22, L490-492)